# Two-step photon up-conversion solar cells

Shigeo Asahi[1], Haruyuki Teranishi[1], Kazuki Kusaki[1], Toshiyuki Kaizu[1] & Takashi Kita[1]

Reducing the transmission loss for below-gap photons is a straightforward way to break the limit of the energy-conversion efficiency of solar cells (SCs). The up-conversion of below-gap photons is very promising for generating additional photocurrent. Here we propose a two-step photon up-conversion SC with a hetero-interface comprising different bandgaps of $Al_{0.3}Ga_{0.7}As$ and GaAs. The below-gap photons for $Al_{0.3}Ga_{0.7}As$ excite GaAs and generate electrons at the hetero-interface. The accumulated electrons at the hetero-interface are pumped upwards into the $Al_{0.3}Ga_{0.7}As$ barrier by below-gap photons for GaAs. Efficient two-step photon up-conversion is achieved by introducing InAs quantum dots at the hetero-interface. We observe not only a dramatic increase in the additional photocurrent, which exceeds the reported values by approximately two orders of magnitude, but also an increase in the photovoltage. These results suggest that the two-step photon up-conversion SC has a high potential for implementation in the next-generation high-efficiency SCs.

[1] Department of Electrical and Electronic Engineering, Graduate School of Engineering, Kobe University, 1-1 Rokkodai, Nada, Kobe 657-8501, Japan. Correspondence and requests for materials should be addressed to S.A. (email: asahis@people.kobe-u.ac.jp).

High-efficiency photovoltaics using n-i-p semiconductor solar cells (SCs) are very promising for generating electrical power by utilizing solar radiation. The conversion efficiency of single-junction SCs is limited to ∼30% of the so-called Shockley–Queisser limit owing to unavoidable losses, such as transmission loss, thermalization loss, Carnot loss, Boltzmann loss and emission loss[1,2]. In particular, the main factors influencing this efficiency limitation are the transmission loss of below-gap photons and the thermalization of photo-generated carriers towards the band edge[2]. Below-bandgap photons with energy smaller than the bandgap of SC are not absorbed and do not contribute to create carriers. Many efforts have been made to realize high-efficiency SCs by breaking the conversion limit and several concepts have been proposed to improve the efficiency[3–9]. One promising SC is the intermediate-band SC (IBSC) containing an additional parallel diode connection, which can reduce the transmission loss[5,6]. The IBSC includes intermediate states in the bandgap. By absorbing a below-gap photon, an electron transits from the valence band (VB) to the intermediate band (IB). Upon absorbing another below-gap photon, the electron is further excited into the conduction band (CB). This two-step photon up-conversion (TPU) process following the absorption of two below-gap photons produces additional photocurrent without degrading the photovoltage. According to ideal theoretical predictions, the IBSC is expected to exhibit extremely high conversion efficiency, >60%, under the maximum concentration and 48.2% under one-sun irradiation[5]. Substantial progress has been made in this field[10–27] since Luque and Martí have proposed this concept of IBSC in 1997 (ref. 5). Generally, the absorption strength of the intraband transition from the IB to the CB is very weak[14–16] and the energy relaxation of the excited electrons into the IB is fast[17,18]. Therefore, improving the second-excitation efficiency in the TPU process strongly influences the conversion efficiency. The optical selection rule for light irradiating the SC surface is relaxed by designing the electronic properties of the quantized states in low-dimensional structures, such as quantum dots (QDs)[19] and impurities[20]. Obviously, carriers in the IB that have long lifetimes have a greater capacity to improve the TPU efficiency because the absorption coefficient between the CB and IB is proportional to the electron density in the initial state of the intraband transition. However, the application of an additional infrared (IR) light corresponding to 40 suns has been observed to improve the external quantum efficiency (EQE) by <0.5% (refs 21,22). The further enhancement of TPU is essential to accomplish high conversion efficiency above 50% under sunlight concentration.

TPU has been known to occur at the hetero-interfaces between III and V semiconductors. Extensive studies have been conducted on photoluminescence up-conversion phenomena[28–34]. Recently, Sellers et al. have proposed a SC structure which attempts optical up-conversion in electrically isolated up-conversion layers[35,36], where high-energy photons emitted by radiative recombination of up-converted electron and hole in the up-conversion layers are absorbed in a SC stacked on it.

Here, we propose a TPU-SC with a hetero-interface, where up-converted electrons are directly collected by the top electrode. We demonstrate an enhancement of the photovoltage as well as a dramatic increase in the photocurrent. This enhancement indicates that the quasi-Fermi gap widens according to the electron excitation into $Al_{0.3}Ga_{0.7}As$.

## Results

**Concept of two-step photon up-conversion solar cell.** We propose a simple structure with a hetero-interface that demonstrates the concept of the TPU-SC. Here, the TPU is effectively realized at a hetero-interface comprising different bandgaps of III-V semiconductors instead of one IB in the bandgap. Figure 1a,b illustrate the schematic band diagram of the proposed TPU-SC with a diode structure of n-$Al_{0.3}Ga_{0.7}As$/$Al_{0.3}Ga_{0.7}As$/ GaAs/p-GaAs on a $p^+$-GaAs(001) substrate. A single InAs QD layer capped by 10 nm GaAs was inserted just below the $Al_{0.3}Ga_{0.7}As$ layer to improve the TPU efficiency. Detailed device structure is described in the Methods section. Here, sunlight irradiates the n-$Al_{0.3}Ga_{0.7}As$ side (left-hand side of Fig. 1a). High-energy photons are absorbed in the $Al_{0.3}Ga_{0.7}As$ layer, and excited electrons and holes drift in opposite directions towards n-$Al_{0.3}Ga_{0.7}As$ and p-GaAs, respectively. The excited electrons reach the n-$Al_{0.3}Ga_{0.7}As$ layer without being captured. The VB discontinuity of ∼170 meV (ref. 37) between $Al_{0.3}Ga_{0.7}As$ and GaAs corresponds to the energy loss that occurs at the hetero-interface. This loss must be carefully designed when optimizing the SC structure. Below-gap photons passing through $Al_{0.3}Ga_{0.7}As$ excite the InAs QDs and GaAs. The excited electrons and holes in GaAs drift in opposite directions in the internal electric field. Although the excited holes can reach the p-GaAs contact layer, the electrons are accumulated at the $Al_{0.3}Ga_{0.7}As$/GaAs interface. Besides, similar spatial carrier separation occurs for photoexcited carriers in the InAs QDs. Electrons accumulated at the hetero-interface are separated from the holes and are expected to exhibit extended lifetimes, which can be in the order of milliseconds in some cases[38]. Long-lived electrons improve the absorption strength for below-gap photons

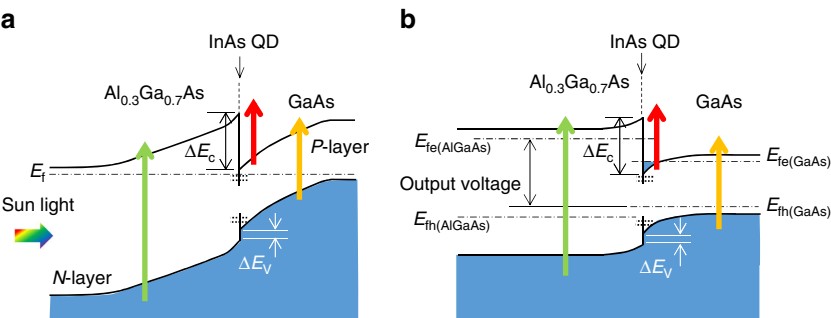

**Figure 1 | Schematic band diagram of TPU-SC.** Diagrams (**a**) at the short-circuit condition and (**b**) at an operating condition. Sunlight irradiates the $Al_{0.3}Ga_{0.7}As$ side. High-energy photons are absorbed in $Al_{0.3}Ga_{0.7}As$, and excited electrons and holes drift in opposite directions towards n-layer and p-layer, respectively. Below-gap photons for $Al_{0.3}Ga_{0.7}As$ excite InAs QDs and GaAs. Long-lived electrons separated from holes are accumulated at the $Al_{0.3}Ga_{0.7}As$/GaAs hetero-interface, inducing a dramatic increase in the two-step photon up-conversion current. $E_f$ in **a** is the Fermi level. $E_{fe}$ and $E_{fh}$ in **b** indicate the quasi-Fermi levels of electrons and holes, respectively. $\Delta E_c$ and $\Delta E_v$ are the conduction band and VB discontinuity, respectively. Quantized states in InAs/GaAs QDs are drawn by horizontal dashed lines.

for GaAs and are efficiently pumped upwards into the $Al_{0.3}Ga_{0.7}As$ barrier. As shown in Fig. 1b, the output voltage of TPU-SC corresponds to the gap of the quasi-Fermi levels for electrons in $Al_{0.3}Ga_{0.7}As$ and holes in GaAs at an operating condition.

**Measurement of external quantum efficiency**. To demonstrate the TPU effect, we measured the EQE and its change ($\Delta$EQE) that were produced by irradiating SC with IR light with photon energy lower than the fundamental edge of InAs QDs. All the measurements were performed at room temperature (290 K). Fig. 2a,b show the EQE and $\Delta$EQE spectra for TPU-SC with InAs QDs, respectively. Without the IR illumination, two clear absorption edges appear at 685 and 875 nm in the EQE spectrum (black colour in Fig. 2a); these edges correspond to the bandgaps of $Al_{0.3}Ga_{0.7}As$ and GaAs, respectively. The EQE signal for high-energy photons (above the bandgap of $Al_{0.3}Ga_{0.7}As$) decreases because of the shallow penetration of the incident light and significant carrier recombination at the surface, which are suppressed by introducing a wider-gap window layer[39]. When excited above the bandgap of $Al_{0.3}Ga_{0.7}As$, the excited electrons and holes are collected at the corresponding electrodes. However, the behaviour of carriers generated by below-gap photons in $Al_{0.3}Ga_{0.7}As$ is different. Below-gap photons are predominantly absorbed in i-GaAs and generate electrons and holes. The excited holes drift towards the p-layer of GaAs. On the other hand, the drift current of excited electrons is partially obstructed at the $Al_{0.3}Ga_{0.7}As$/GaAs interface, resulting in a significant drop of the EQE signal below the bandgap of $Al_{0.3}Ga_{0.7}As$. The observed photocurrent in the wavelength region between the bandgaps of $Al_{0.3}Ga_{0.7}As$ and GaAs is caused by the thermal and tunnelling escape of the accumulated electrons at room temperature. In this wavelength region, the EQE signal also shows a gradual decrease with increasing wavelength because the optical absorption coefficient becomes small with increasing wavelength. In the near-IR wavelength region below the bandgap of GaAs, the EQE signal decreases drastically and shows a small structure at 912 nm that can be attributed to thermally excited carriers from the deep quantized states of the InAs wetting layer[21,40].

Figure 3a shows the temperature dependent EQE spectra. At low temperature, the absorption edges of $Al_{0.3}Ga_{0.7}As$ and GaAs are relatively steep owing to the excitonic feature. With increasing the temperature, the absorption edges shift and the below-gap state attributed to the InAs-wetting layer appears gradually. The EQE signal from QDs was very weak and below the detection limit because of the deeper quantized state. Figure 3b shows the temperature dependence of the current density at 780 nm excited by a laser diode (LD). 780 nm photons directly excite i-GaAs. The excitation power density was 47 mW cm$^{-2}$. The current density increases with increasing the temperature. The inset of Fig. 3b indicates the applied bias voltage dependence of the estimated thermal activation energy $E_A$. $E_A$ monotonically decreases with increasing the electric field because of lowering the effective barrier height at the hetero-interface. $E_A$ shows the maximum of $221 \pm 3$ meV at 0.02 V. Conversely, applying higher positive bias voltage weakens the internal electric field significantly and makes flatter the band. As the forward current increases even at the same bias condition with increasing the temperature, the detected photocurrent decreases rapidly with flatten the band. Thereby, $E_A$ decreases and finally becomes negative with increasing the bias voltage. The maximum $E_A$ excellently coincides with the estimated CB discontinuity between $Al_{0.3}Ga_{0.7}As$ and GaAs[37]. Figure 3c shows the temperature dependence of the current density at 912 nm corresponding to the wetting layer state. We recorded the current at the bias of 0.02 V. Here, the excitation light was produced by a supercontinuum laser, passed through a 270 mm single monochromator. The monochromatic excitation-laser line width was 9.6 nm. The EQE line width of the wetting layer state is ~15 nm and the temperature drift of the wetting layer state is ~2.9 nm, so that we fixed the excitation wavelength in this experiment. The evaluated thermal activation energy was $254 \pm 5$ meV. Photo-excited electrons are thermally excited from the GaAs edge to the $Al_{0.3}Ga_{0.7}As$ one, from the InAs wetting layer state to the $Al_{0.3}Ga_{0.7}As$ edge, and from the ground state of the QD transition to the GaAs edge. We did not confirm an obvious change caused by thermal excitation of holes, suggesting photo-excited holes reach the p-GaAs contact without captured at the hetero-interface. These optical responses are linear with the excitation density, and we did not observe any obvious nonlinear two-photon absorptions, as discussed later.

The temperature dependence of the photoluminescence (PL) intensity reflects the change in the recombining carrier density in QDs. That change is caused by the thermal carrier escape from the confined state. Figure 4a,b show PL spectra for TPU-SC with InAs QDs measured at various temperatures and the temperature dependence of the integrated PL intensity, respectively. The wavelength and power density of the excitation laser were 784 nm and 2.1 mW cm$^{-2}$, respectively. The PL represents the ground state transition of InAs QDs. The PL peak shifts with temperature and obeys a well-known Varshni's relationship. The integrated PL intensity decreases with increasing the temperature. The thermal activation energy evaluated from the Arrhenius plot is $244 \pm 4$ meV, which coincides with the CB discontinuity between the ground state of InAs QD and the GaAs band edge[41]. Figure 5 summarizes these results we obtained. This clear picture clarifies available energy states for confined carriers at the hetero-interface of TPU-SC with InAs QDs.

Next, we discuss the EQE spectrum measured under illumination by an IR-LD. The wavelength of the IR-LD was 1,300 nm, which is sufficiently long to prevent interband transitions and can only induce intraband transitions, as shown in Fig. 4a. In this case, the spectrum was dramatically changed. It must be noted that the EQE signal (drawn by magenta colour in Fig. 2a) increases remarkably in the wavelength region between the bandgaps of $Al_{0.3}Ga_{0.7}As$ and GaAs and seems to eliminate the

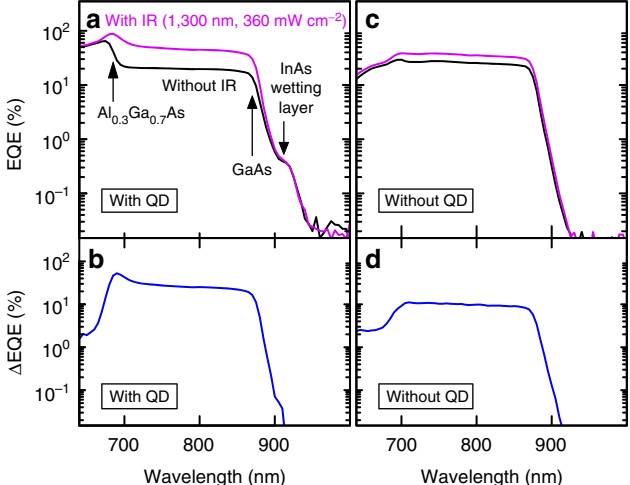

**Figure 2 | EQE spectra obtained with and without IR light and $\Delta$EQE spectra measured at 290 K.** (a,c) show EQE spectra for TPU-SC with and without InAs QDs, respectively. The black and magenta lines represent the EQE spectra measured with and without 1,300 nm LD illumination, respectively. (b,d) show $\Delta$EQE spectra of TPU-SC with and without InAs QDs, respectively. $\Delta$EQE is defined as the difference between the EQE signals measured with and without the 1,300 nm LD illumination.

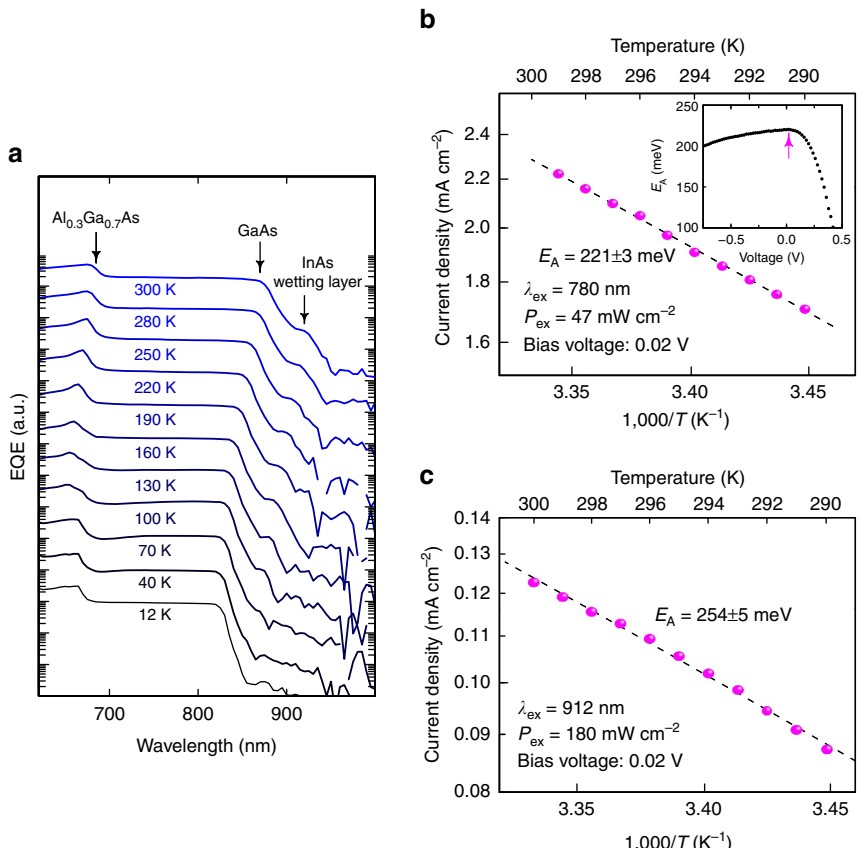

**Figure 3 | EQE spectra measured at various temperatures and the temperature dependence of the current density.** (**a**) EQE spectra for TPU-SC with InAs QDs at various temperatures. (**b,c**) show the temperature dependences of the current density at 780 and 912 nm, respectively. 780 and 912 nm photons directly excite i-GaAs and the InAs-wetting layer state, respectively. Magenta circles indicate the measured current density at the bias voltage of 0.02 V as a function of the temperature. The dashed line represents the result of the Arrhenius-type fitting. $E_A$ is the estimated thermal activation energy.

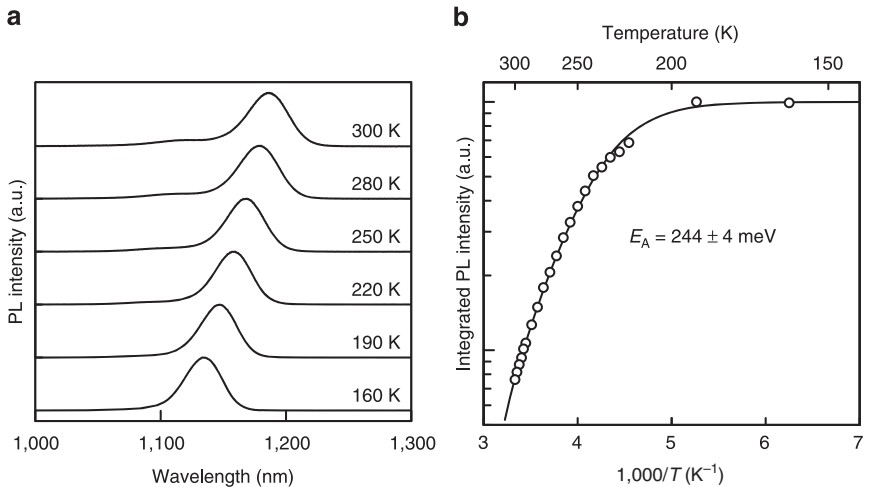

**Figure 4 | Photoluminescence spectrum for TPU-SC with InAs QDs.** (**a**) Photoluminescence spectra measured at various temperatures and (**b**) the temperature dependence of the integrated photoluminescence intensity.

loss caused by the electron accumulation at the hetero-interface. We defined ΔEQE (blue colour in Fig. 2b) as the difference between the EQE obtained with and without the 1,300 nm LD illumination. The ΔEQE enhancement was ∼30%. We also observed an increase in the EQE signal attributed to the quantized states of the InAs wetting layer. However, ΔEQE at the InAs QD ground state of 1,186 nm was very weak, suggesting that optical absorption in the single InAs QD layer with the in-plane QD

density of $\sim 1.0 \times 10^{10}\,cm^{-3}$ is not enough to contribute to the change in the current generation at the QD ground state. Most of the excited electrons in GaAs were separated from the holes and accumulated at the $Al_{0.3}Ga_{0.7}As/GaAs$ interface. Such densely accumulated long-lived electrons are easily pumped into the $Al_{0.3}Ga_{0.7}As$ barrier by the 1,300 nm LD light, which accomplishes efficient TPU at the hetero-interface. Additionally, we fabricated a reference TPU-SC without InAs QDs. The EQE

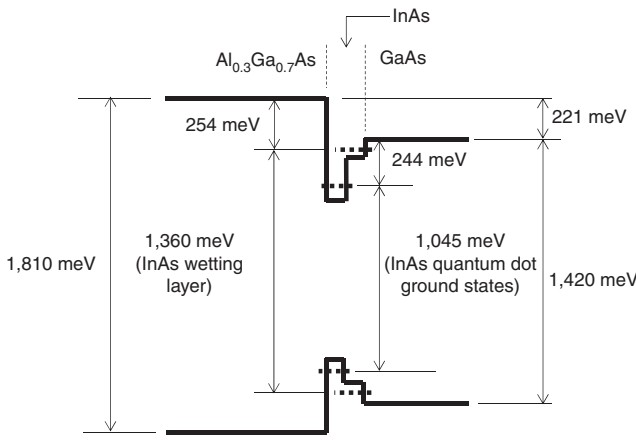

**Figure 5 | Energy states for confined carriers at the hetero-interface of TPU-SC with InAs QDs at approximately 300 K.** Interband transition energies are determined by the EQE and photoluminescence spectra at 300 K. Energy differences in the conduction band lineup are evaluated by the temperature dependence of the photocurrent density in Fig. 3 and the photoluminescence intensity in Fig. 4.

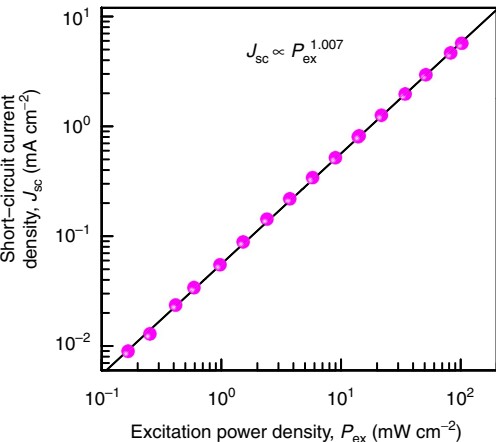

**Figure 6 | Excitation power dependence of short-circuit current density $J_{sc}$ of TPU-SC with InAs QDs when excited by a 780 nm LD.** The 780 nm photons traverse $Al_{0.3}Ga_{0.7}As$ and excite the i-GaAs layer directly. The open circles indicate the experimental results at 297 K. The solid line represents a fitting line created by the relation $J_{sc} \propto P_{ex}^n$, where $P_{ex}$ is the excitation power density.

and $\Delta EQE$ spectra for the reference TPU-SC are shown in Fig. 2c,d, respectively. The same absorption edges of $Al_{0.3}Ga_{0.7}As$ and GaAs appear in the EQE spectrum. As shown in Fig. 2a,c, the EQE drop observed below the bandgap of $Al_{0.3}Ga_{0.7}As$ was significant for TPU-SC with InAs QDs, which is caused by extra carrier recombination in QDs. As shown in Fig. 2d, $\Delta EQE$ is obviously generated even in TPU-SC with the hetero-interface of $Al_{0.3}Ga_{0.7}As$/GaAs without InAs QDs. Comparison between the $\Delta EQE$ spectra suggests that the hetero-interface containing InAs QDs improves the TPU efficiency for the accumulated electrons. The optical selection rule of the intersubband transition of electrons in an ideal two-dimensional structure is forbidden for light irradiating the two-dimensional plane perpendicularly[42]. The finite thickness of the accumulation layer relaxes the selection rule, and, moreover, InAs QDs play a role enhancing the TPU efficiency. Generally, it is well known that the electronic wavefunctions in QDs are quantized on all three dimensions, and light of all polarization directions induces intersubband transitions[43]. Thus, electrons at the hetero-interface obey the selection rule modified by QDs and are efficiently pumped into the CB of $Al_{0.3}Ga_{0.7}As$ by the 1,300 nm LD illumination.

**Short-circuit current generated by 780 nm photo-excitation.** Figure 6 shows the short-circuit current density of TPU-SC with InAs QDs as a function of the excitation power density of a single-colour excitation light source. We used a 780 nm LD for excitation. The 780 nm photons traversed $Al_{0.3}Ga_{0.7}As$ and directly excited the intrinsic layer of GaAs. The excited electrons drifted towards the n-layer and were obstructed at the hetero-interface; subsequently, they were partially extracted by thermal and tunnelling processes at the interface and finally reached the n-side electrode, generating a photocurrent. The short-circuit current density clearly exhibits a linear dependence on the excitation power density, indicating that no nonlinear two-photon absorption occurs in $Al_{0.3}Ga_{0.7}As$.

**TPU at biased conditions.** Next, we studied TPU phenomena at biased conditions using two-colour photo-excitations. Figure 7a shows typical current–voltage curves obtained for the TPU-SC with InAs QDs with illumination from the 780 nm LD and the additional 1,300 nm LD. The excitation power density of the 780 nm LD was 110 mW cm$^{-2}$. When only the 1,300 nm LD was

used for the excitation, no changes were observed in the photocurrent and photovoltage, indicating that the below-gap photons for GaAs do not cause non-linear two-photon absorption in GaAs. When irradiated by the 780 nm LD, the TPU-SC produces both photocurrent and photovoltage and the 780 nm photons traverse $Al_{0.3}Ga_{0.7}As$ and excite GaAs. The excited electrons drift towards n-$Al_{0.3}Ga_{0.7}As$ and are obstructed at the hetero-interface. The accumulated electrons at the hetero-interface are partially extracted by thermal and tunnelling processes and thus generate electric power. By adding the 1,300 nm LD illumination, we observed an obvious enhancement in the photocurrent; for a density of 320 mW cm$^{-2}$, the photocurrent increased by 0.6 mA cm$^{-2}$. This value is rather high and approximately two orders of magnitude greater than previously reported values, as described in the Discussion section. Generally, the intraband excitation strength is proportional to the electron density in the initial state. Because of the carrier separation in the internal electric field, extremely long-lived electrons are densely accumulated at the hetero-interface and fill all the confinement states of the InAs QDs and the wetting layer. Here, it must be noted that we confirmed an increase in the photovoltage by adding the 1,300 nm LD illumination. This demonstrates that TPU enhances quasi-Fermi level splitting, which is a key feature that characterizes the operation of the TPU-SC. When irradiated by the 780 nm LD, the SC operates only in the GaAs region, and the open-circuit voltage is predominantly limited by GaAs. TPU populates electrons in $Al_{0.3}Ga_{0.7}As$ and consequently, the quasi-Fermi levels split further.

Figure 7b,c summarize the 1,300 nm excitation power dependence of the change in the short-circuit current density, $\Delta J_{sc}$, and the open-circuit voltage, $\Delta V_{oc}$. $\Delta J_{sc}$ and $\Delta V_{oc}$ exhibit different behaviours according to a model proposed in the Methods section. $\Delta J_{sc}$ is proportional to $P_{ex}^{0.73}$, where $P_{ex}$ is the 1,300 nm excitation power density. Generally, the short-circuit current has a linear relationship with the excitation density. The measured $\Delta J_{sc}$ shows a sub-linear response. Figure 7d indicates the dependence of the evaluated $n$ value on the reverse-bias voltage in the relationship $J_{sc} \propto P_{ex}^n$. As the reverse-bias voltage increases, $n$ increases and approaches unity. The dense space charge accumulated at the hetero-interface weakens the electric field, resulting in a sub-linear response to the excitation density because a stronger electric field improves the carrier collection efficiency

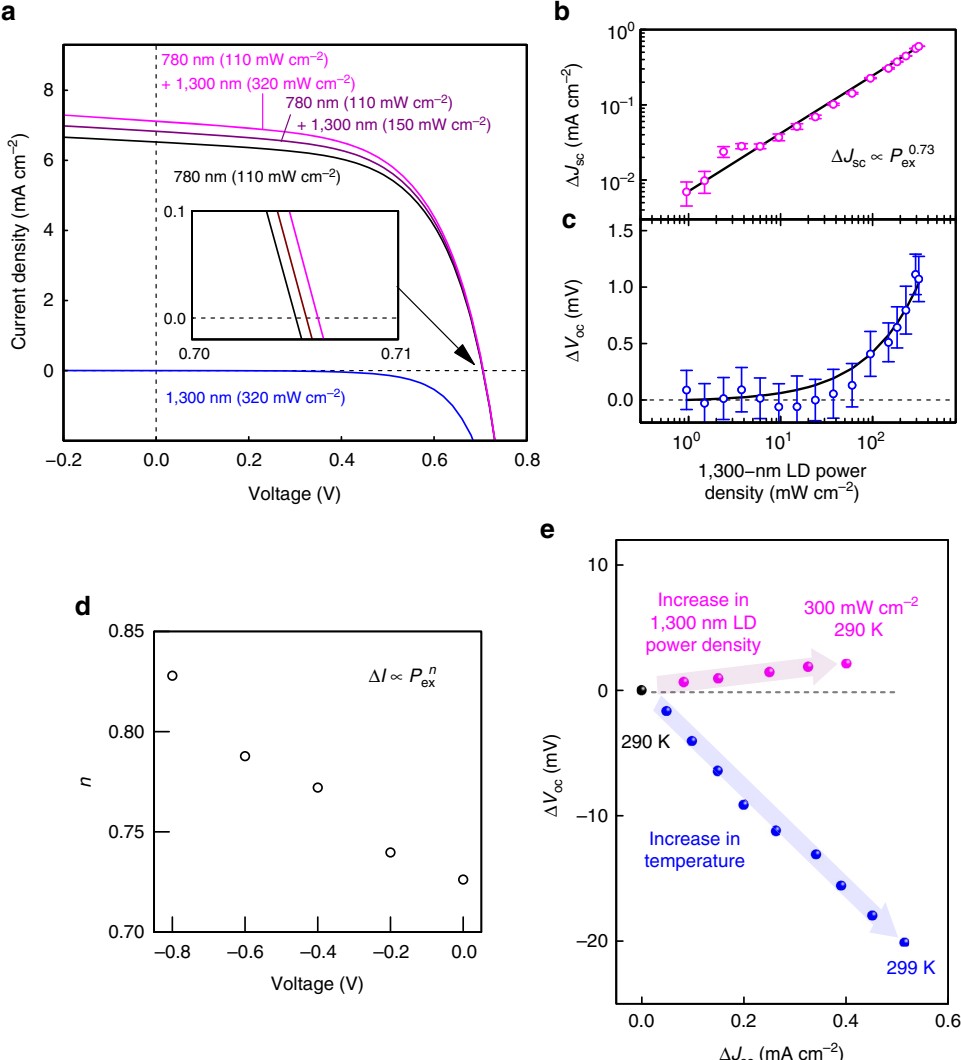

**Figure 7 | Two-step photon up-conversion current at biased conditions.** (**a**) Current-voltage curve obtained with light illumination. The black and blue lines correspond to irradiation by a 780 nm LD and a 1,300 nm LD, respectively. The magenta line indicates the result obtained with irradiation by both the 780 nm and 1,300 nm LDs. The inset shows the magnification of the open-circuit voltage. All light sources operated in the continuous-wave mode without any optical choppers. Dependence of up-converted characteristics of (**b**) the short-circuit current ($\Delta J_{sc}$) and (**c**) open-circuit voltage ($\Delta V_{oc}$) on the 1,300 nm LD excitation power. The solid lines indicate fitted curves determined by a model proposed in the Methods section. The error bars represent the standard error in the measurements. (**d**) Dependence of evaluated $n$ value on the reverse-bias voltage according to the relationship, $\Delta J_{sc} \propto P_{ex}^{n}$, where $P_{ex}$ is the 1,300 nm excitation power density. (**e**) Comparison of the $\Delta J_{sc}$-$\Delta V_{oc}$ relation between increasing the 1,300 nm LD power density and increasing the temperature. $\Delta J_{sc}$ and $\Delta V_{oc}$ in Fig. 7e indicate the difference in $J_{sc}$ and $V_{oc}$ against the values (black circle) measured by a 780 nm LD with the intensity of 47 mW cm$^{-2}$ at 290 K.

of the TPU. Conversely, $\Delta V_{oc}$ increases non-linearly with the 1,300 nm -excitation power density. A detailed model that reproduces $\Delta J_{sc}$ and $\Delta V_{oc}$ is proposed and discussed in the Methods section. As given in equation (4), $\Delta V_{oc}$ is an increase against $V_{oc,single}$ which is the open-circuit voltage measured at the single-colour excitation without the 1,300 nm LD illumination. The increase in $\Delta V_{oc}$ includes effect of the voltage boost effect at the hetero-interface, which follows an increase in the extra photocurrent $\Delta J_{sc}$ created by the additional 1,300 nm LD illumination. Next, we demonstrated a difference between the TPU effect caused by the optical process and the thermal excitation effect. To confirm the contribution of 1,300 nm LD illumination to $\Delta V_{oc}$, we carefully measured $\Delta V_{oc}$ as a function of $J_{sc}$ controlled by the 1,300 nm LD illumination or temperature. The results are summarized in Fig. 7e. The blue circles indicate $\Delta V_{oc}$ recorded by changing $J_{sc}$ controlled by temperature. Here, the 1,300 nm LD does not shine the device. With increasing the

temperature, $J_{sc}$ increases because of increasing thermal carrier excitation, and, resultantly, $\Delta V_{oc}$ reduces. This is a well-known phenomenon. As the bandgap change in this temperature variation is ~4.5 meV which is given by $5 \times 10^{-4}$ eV K$^{-1}$ of the temperature dependence of the bandgap of GaAs, the observed change in $\Delta V_{oc}$ was almost caused by the thermal carrier excitation effect. Conversely, when the 1,300 nm LD with the excitation power density of 300 mW cm$^{-2}$ illuminates the device at 290 K, $\Delta V_{oc}$ slightly increases, despite increasing $J_{sc}$ similarly. The clear difference between the thermal effect and the TPU by the second-photon flux proves the concept of the proposed TPU-SC.

**Theoretical prediction of the conversion efficiency.** Here, we estimate the expected conversion efficiency of the TPU-SC, using the detailed balance framework which considers a steady state

between carrier generation and recombination at the optimum operation point of SC[1,25]. We ignore nonradiative processes in SC for predicting the ideal limit of the conversion efficiency. As shown in Fig. 2d, TPU itself occurs at the $Al_{0.3}Ga_{0.7}As$/GaAs hetero-interface, and InAs QDs play a role enhancing the TPU efficiency. Here, we neglected QD states enhancing the TPU efficiency in our calculation, and we simply assumed a perfect TPU at the hetero-interface. In this calculation, we consider that solar radiation is a black body with a temperature of 6,000 K and the temperature of the TPU-SC is 300 K. The calculation model assumes an absorptivity of 1 and good photon selectivity[25]. We maintained the bandgap energy (1.80 eV) of $Al_{0.3}Ga_{0.7}As$ fixed and varied the CB offset ($E_2$) and the VB offset ($\Delta E$). The TPU of electrons occurs at the potential discontinuity of $E_2$. Figure 8 shows the calculated results as a function of $E_2$ at a solar concentration of 1 and 1,000 suns. Increasing the $E_2$ with a fixed $\Delta E$ results in a decreasing $E_1$ (see inset of Fig. 8). When $E_2$ and $\Delta E$ are zero, the conversion efficiency coincides with that of a single-junction $Al_{0.3}Ga_{0.7}As$ SC. Regardless of $\Delta E$, the conversion efficiency increases as $E_2$ increases because the contribution of the TPU increases, namely, the transmission loss decreases. Finally, the efficiency reaches a peak; under the one-sun irradiation, the maximum conversion efficiency is 44% at $E_2 = 0.63$ eV when $\Delta E = 0$. This value coincides with the efficiency calculated for the well-known ideal IBSC[5]. Furthermore, under 1,000-sun irradiation, the conversion efficiency exceeds 50%, even at $\Delta E = 0.2$ eV. The $E_2$ at which the conversion efficiency exhibits a peak shifts with varying $\Delta E$. Increasing the $\Delta E$ leads to a monotonic decrease in the conversion efficiency, which is caused by a voltage loss at the hetero-interface. These results suggest that a zero VB discontinuity ($\Delta E = 0$) achieves maximum conversion efficiency. To reduce $\Delta E$, the use of other material systems showing a type-II band alignment would accentuate the intrinsic nature of the TPU-SC.

## Discussion

Intraband transition is recognized as a very weak phenomenon that depends on the carrier density in the initial state. Here, we observed a strong TPU when using a hetero-structure that included InAs QDs. Considering the results of $\Delta J_{sc}$ shown in Fig. 7b, we estimated the intraband absorption coefficients for the

TPU at the hetero-structure. We assumed that $\Delta J_{sc}$ and the absorption coefficient obey the following relationship that is based on the Beer–Lambert law:

$$\Delta J_{sc} = qN_{in}\{1 - \exp(-\alpha d)\}. \qquad (1)$$

where $q$ is the elementary charge, $N_{in}$ is the incident 1,300 nm photon flux, $\alpha$ is the absorption coefficient and $d$ is the thickness of absorber. When calculating $N_{in}$, we considered a reflectivity of 29.8% at the SC surface. The maximum value of $\alpha d$ estimated from equation (1) was $9 \times 10^{-3}$, depending on the 1,300 nm excitation power density. $d$ corresponds to the thickness of the electrons gas forming around the hetero-interface. Roughly assuming that the electron gas concentrates in the InAs QDs layer, $d$ is considered to be the InAs-QD height of 3 nm; consequently, the maximum $\alpha$ becomes 30,000 cm$^{-1}$. This value is rather high compared to coefficients previously reported in the literature, which were in the range 400–2,000 cm$^{-1}$ (refs 18,26,27). The developed TPU-SC presents extremely long-lived electrons accumulated at the hetero-interface that fill all the confinement states of the InAs QDs and wetting layer, leading to this strong intraband excitation. The additional 1,300 nm LD produces not only a dramatic increase in the photocurrent, which is two orders of magnitude greater than ever observed, but also an increase in the photovoltage. The typical illumination power density used in the experiment corresponded to approximately 17 suns. This concentration ratio is relatively low, demonstrating the efficient TPU effect, and is expected to be easily realized.

Next, we compare $\Delta J_{sc}$ with results reported in several references. As shown in Fig. 7a, the maximum $\Delta J_{sc}$ of our TPU-SC was 0.6 mA cm$^{-2}$ at the additional 1,300 nm IR-LD power density of 320 mW cm$^{-2}$. In ref. 14, two-step photon absorption in GaSb/GaAs type-II QD-IBSC has been reported, where the maximum $\Delta J_{sc}$ obtained by irradiating additional IR light with the intensity of 750 W cm$^{-2}$ was estimated $\sim 10$ nA cm$^{-2}$ at 200 K and became smaller than 1 nA cm$^{-2}$ at the temperature above 250 K. In ref. 38, we reported a saturable behaviour of $\Delta J_{sc}$ in IBSC including InAs QDs embedded in $Al_{0.3}Ga_{0.7}As$/GaAs quantum well. In that study, the maximum $\Delta J_{sc}$ was 0.15 µA cm$^{-2}$ when excited by the additional IR light with the power density of 56 mW cm$^{-2}$. As the saturated $\Delta J_{sc}$ is proportional to the IR power density, $\Delta J_{sc}$ can be estimated to be 0.86 µA cm$^{-2}$ at the IR power density of 320 mW cm$^{-2}$ used in our experiment. Elborg

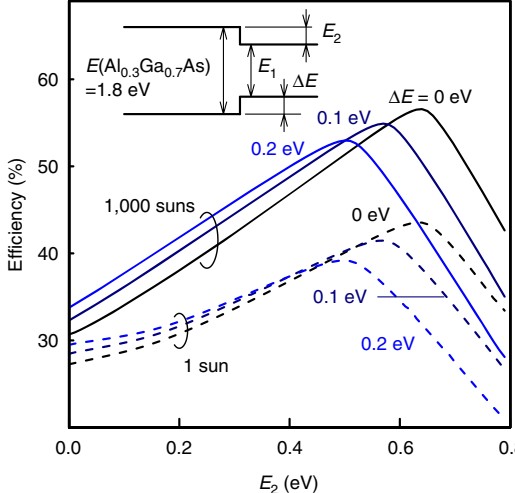

**Figure 8 | Detailed balance calculation of TPU-SC.** The horizontal axis represents the conduction band offset ($E_2$), shown in the inset. The bandgap energy of the host material ($Al_{0.3}Ga_{0.7}As$) is fixed at 1.80 eV. The solid and dashed lines correspond to the results at a solar concentration of 1 and 1,000 suns, respectively. $\Delta E$ is the VB offset (inset).

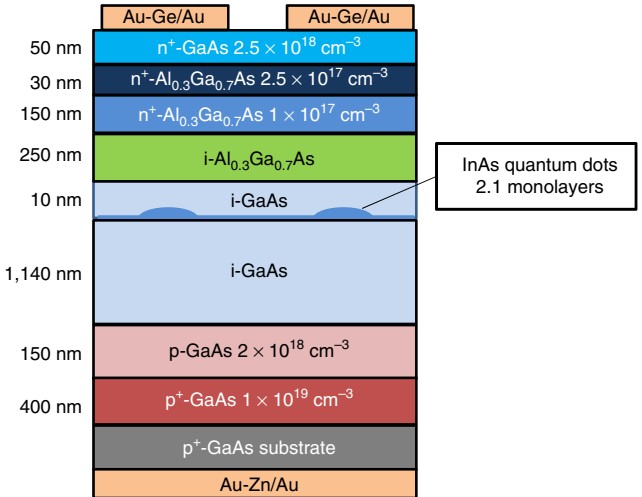

**Figure 9 | Schematic of the structure of the TPU-SC.** TPU-SC was fabricated using solid-source molecular beam epitaxy. The intrinsic layer comprises AlGaAs/GaAs. InAs QDs are inserted at the hetero-interface.

*et al.* investigated the voltage dependence of extra photocurrent for the GaAs/Al$_{0.28}$Ga$_{0.72}$As QD-IBSC[23]. In that literature, $\Delta J_{sc}$ was 0.44 µA cm$^{-2}$ when excited at the additional IR power density of 1,400 mW cm$^{-2}$. Sellers *et al.*[24] investigated InAs/GaAs QD-IBSC inserting GaP strain-balancing layer between each QD layer. Here, the maximum $\Delta J_{sc}$ was 3 µA cm$^{-2}$ at the additional IR power density of 300 mW cm$^{-2}$. Thus, $\Delta J_{sc}$ of our TPU-SC exceeded all the reported values by greater than two orders of magnitude.

The SC device used in this study was designed and fabricated to demonstrate a strong TPU phenomenon. However, the SC structure presented in this paper is not the final version of this device. A detailed optimized design considering the theoretical predictions shown in Fig. 8 is necessary to achieve the best performance of the TPU-SC. For example, the position of the hetero-interface becomes important. Furthermore, as the cause for TPU reduces when the electric field is strongly reduced, we need to perform detailed simulations of the band profile at the operating point in order to maintain a moderate internal electric field even at the operating point by controlling the doping profile near the hetero-interface. In any case, the obtained results suggest that the SC structure proposed in this paper has a high potential for implementation in the next-generation high-efficiency SCs.

## Methods

**Solar cell fabrication.** The TPU-SC was fabricated on a $p^{+}$-GaAs (001) substrate using solid-source molecular beam epitaxy. The detailed structure is illustrated in Fig. 9. A 150-nm-thick p-GaAs (Be: $2 \times 10^{18}$ cm$^{-3}$) layer was grown over a 400-nm-thick $p^{+}$-GaAs (Be: $1 \times 10^{19}$ cm$^{-3}$) buffer layer at a substrate temperature of 550 °C. The substrate temperature was monitored during the growth using an infrared pyrometer. Subsequently, an i layer with the structure Al$_{0.3}$Ga$_{0.7}$As (250 nm)/GaAs (10 nm)/InAs QDs/GaAs (1,140 nm) was deposited. The nominal thickness of InAs was 0.64 nm (2.1 monolayers). The typical height and width of the QDs was 3 and 20 nm, respectively, and the QD density was approximately $1.0 \times 10^{10}$ cm$^{-3}$. The substrate temperature before the deposition of the InAs QDs was 550 °C. The InAs QDs and the subsequent 10-nm-thick GaAs capping layer were grown at 490 °C. The thin GaAs capping layer maintained the optical quality of the InAs QDs, even if Al$_{0.3}$Ga$_{0.7}$As is grown at 490 °C, which is lower than the optimum growth temperature. Finally, n$^{+}$-GaAs (Si: $2.5 \times 10^{18}$ cm$^{-3}$), n$^{+}$-Al$_{0.3}$Ga$_{0.7}$As (Si: $2.5 \times 10^{17}$ cm$^{-3}$), and n-Al$_{0.3}$Ga$_{0.7}$As (Si: $1 \times 10^{17}$ cm$^{-3}$) layers were grown on the SC structure at a substrate temperature of 500 °C. The beam-equivalent pressure of the As$_2$ flux was $1.15 \times 10^{-3}$ Pa. Then, metal Au/Au-Ge and Au/Au-Zn contacts were created on the top and the bottom surfaces, respectively. The dimensions of the SC were $4 \times 4$ mm$^2$. Note that the SC structure used in this study was not optimized for obtaining a high conversion efficiency according to the theoretical work shown in Fig. 8 but was fabricated to demonstrate the fundamental TPU effects on the SC characteristics. Further development, such as the optimization of the thickness and doping concentration of each layer as well as introduction of a window layer and anti-reflection coating, is required to obtain the best performance.

**EQE and ΔEQE measurement.** The EQE and ΔEQE measurement was conducted at various temperatures. The excitation light was produced by a tungsten halogen lamp, passed through a 140 mm single monochromator, and chopped by an optical chopper with a frequency of 800 Hz. The excitation power density depended on the wavelength and the integrated power density was approximately 2 mW cm$^{-2}$, much lower than that of the 1 sun solar irradiance of 100 mW cm$^{-2}$. The beam diameter of the monochromatic light was 1.2 mm on the SC surface. The photocurrent was detected by a lock-in amplifier synchronized with the optical chopper. The photocurrent was measured under short-circuit conditions without external bias voltage. Here, the EQE was defined as the efficiency of the photocurrent generation under monochromatic excitation, namely, the number of electrons collected as the photocurrent normalized by the incident photon flux at each wavelength. The TPU was demonstrated by measuring the change in the EQE signal amplitude under two-colour excitation using two types of light sources. The first interband-excitation light source was a monochromated tungsten halogen lamp. The second intraband light source was a continuous-wave LD with the 1,300 nm emission, which was used for pumping electrons accumulated at the Al$_{0.3}$Ga$_{0.7}$As/GaAs interface into the Al$_{0.3}$Ga$_{0.7}$As barrier. The 1,300 nm LD wavelength was sufficiently long to prevent interband transitions. The beam diameter of the 1,300 nm LD was 1.2 mm on the SC surface. The excitation power density of the 1,300 nm LD was 360 mW cm$^{-2}$. ΔEQE was defined as the difference between the EQE obtained with and without the 1,300 nm LD illumination.

**Current–voltage measurement.** In this measurement, we used two LDs. The first interband excitation was obtained using a 780 nm LD; the 780 nm photons traversed Al$_{0.3}$Ga$_{0.7}$As and directly excited the i-GaAs layer. The second intraband-excitation light source was a 1,300 nm LD. Both excitation sources operated in the continuous-wave mode. The excitation density was varied by using a reflective neutral density filter. The beam diameter of the both LD was 1.2 mm on the SC surface. When the bias voltage was applied on the SC, the photocurrent was obtained using a Keithley 2400 source meter. The measurements were performed at 297 K without a temperature controller.

**Modelling of $\Delta J_{sc}$ and $\Delta V_{oc}$.** Generally, the short-circuit current has a linear relationship with the excitation density. However, as described in the Results section, $\Delta J_{sc}$ exhibited a clear sub-linear response to the excitation density; this is because dense space charge accumulated at the hetero-interface weakens the electric field and the carrier collection efficiency of the TPU decreases. Therefore, we modified equation (1) as follows to interpret the results phenomenologically:

$$\Delta J_{sc} = qN_{in}^{n}\{1 - \exp(-\alpha d)\}, \quad (2)$$

where $n$ is a fitting parameter. The result of the reverse-bias-voltage dependence of the evaluated $n$ value shown in Fig. 7d indicates that our hypothesis is realistic.

On the other hand, when modelling $\Delta V_{oc}$, we used the equation describing the $V_{oc}$ of a single-junction SC:

$$V_{oc} = \frac{k_B T}{q}\ln\left(\frac{J_{sc}}{J_0} + 1\right), \quad (3)$$

where $k_B$ is the Boltzmann constant, $T$ is the temperature, and $J_0$ is the saturation current. For two-colour photo-excitation measurements, $V_{oc}$ can be divided into $V_{oc,single}$ and $\Delta V_{oc}$, where $V_{oc,single}$ is the open-circuit voltage at the single-colour excitation. Likewise, $J_{sc}$ can be divided into $J_{sc,single}$ and $\Delta J_{sc}$, where $J_{sc,single}$ is the short-circuit current at the single-colour excitation. By substituting $V_{oc}$ and $J_{sc}$ with $V_{oc,single} + \Delta V_{oc}$ and $J_{sc,single} + \Delta J_{sc}$, $\Delta V_{oc}$ is written as

$$\Delta V_{oc} = \frac{k_B T}{q}\ln\left(\frac{J_{sc,single} + \Delta J_{sc}}{J_0} + 1\right) - V_{oc,single}. \quad (4)$$

Equation (4) was used to fit the results of Fig. 7b,c.

**Data availability.** The data that support the findings of this study are available from the corresponding author upon request.

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

## Acknowledgements

This work was partially supported by the Incorporated Administrative Agency New Energy and Industrial Technology Development Organization (NEDO) and the Japan Society for the Promotion of Science (JSPS) KAKENHI Grant Number JP15K13953. One of the authors (T. Kita) thanks Prof. Masakazu Sugiyama of the University of Tokyo for fruitful discussions.

## Author contributions

S.A. originated the concept, designed and carried out experiments, performed modelling and the data analysis and wrote the manuscript. H.T. and T. Kaizu designed and fabricated the solar devices. K.K. performed the theoretical calculation. T. Kita co-wrote the manuscript and was in charge of overall direction and planning.
