## [Peer Review File · Nature Communications]

Reviewers' comments:

Reviewer #1 (Remarks to the Author):

The article by Shigeo Asahi, "Two-step photon up-conversion solar cells" in its current requires additional refinement to improve its readability:

line 8 and throughout the text - 'transmission loss' term is ambiguous, would 'recombination loss' be more appropriate?

lines 16-17 --- dramatic increase in the photocurrent, which exceeded the reported value by approximately two orders of magnitude, but also an increase in the photovoltage.

'Did the photocurrent or the change in the photocurrent exceed the reported value?'

lines 164 and on:

Theoretical prediction of the conversion efficiency of the TPU-SC.

Here, we estimate the expected conversion efficiency of the TPU-SC, which is calculated in a detailed valance framework.

What is 'detailed valance framework'. Give more details of the model, its assumptions and approximations.

I suggest proofreading it.

Experimental results clearly in this and other articles by Asahi, et. al. indicate that the TPU conversion mechanism is working.

The references may be expanded to include results on InAs/GaAs QD IBSC published by other groups, for example, by R.S. Goldman, S. Forrest from UMich, Ann Arbor; M.F. Doty and J. Zide from University of Delaware.

The authors need to explain how their approach is fundamentally different from the one presented in:

Diane G. Sellers, Stephen J. Polly, Yujun Zhong,
Seth M. Hubbard, Joshua M. O. Zide, Matthew F. Doty,
IEEE JOURNAL OF PHOTOVOLTAICS, VOL. 5, NO. 1, JANUARY 2015
"New Nanostructured Materials for Efficient Photon Upconversion".

The article can be accepted for publication after a detailed revision.

Reviewer #2 (Remarks to the Author):

The manuscript deals with the timely topic of Intermediate Band Solar Cells which represents one of the third generation photovoltaic concepts. This research field is currently stimulating intense work focused on the band gap engineering of solar cell heterostructures, widely recognized by the involved community as fundamental to accomplish the theoretical expectations of improved conversion efficiency. In particular most of the current literature, as the manuscript under review, studies the employment of semiconductor quantum dots to engineer energy bands suitable for below band gap photon absorption.

Within this frame, the present manuscript proposes an heterostructure scheme based on a combination of different semiconductor compounds, namely AlGaAs/GaAs/InAs, claimed to enable an efficient two step photon up conversion(TUP) process, at the base of the Intermediate band solar cell concept. Even though the obtained enhancement in external quantum efficiency of the investigated device is undoubtedly large, I think that a more detailed discussion on this effect is missing. To be clearer, the improvement is claimed in the abstract and in the manuscript as the best "ever reported" but for supporting this claim, only reference 16, from the same authors and using a similar dot in well structure, is cited. For a journal like Nature Communications I would expect a more extensive discussion on such a claim, with respect to a larger suite of literature results. Relevant and more updated works proposing different approaches and analysing the competition between equilibrium and non-equilibrium charge-transfer processes must be cited in this respect:

J Hwang et al, PHYSICAL REVIEW APPLIED 1, 051003 (2014) - Multiphoton Sub-Band-Gap Photoconductivity and Critical Transition Temperature in Type-II GaSb Quantum-Dot Intermediate-Band Solar Cells

M. Gioannini et al., "Simulation of quantum dot solar cells including carrier intersubband dynamics and transport," IEEE J. Photovoltaics 3(4), 1271 (2013).

A. Creti et al, "Role of charge separation on two-step two photon absorption in InAs/GaAs quantum dot intermediate band solar cells", Applied Physics Letters 2016, 108, 063901

A. Varghese et al., "Complete voltage recovery in quantum dot solar cells due to suppression of electron capture", Nanoscale, 2016, 8, 7248

In the introduction, it is stated that the "key factor" at the base of the increase in TPU efficiency is "the spatial potential fluctuations at the heterointerface". It seems to me that this effect is inferred to the specific potential barrier, not to any fluctuations. This point should be clarified.

All along the text, electrons are said to be accumulated at the hetero-interface AlGaAs/GaAs/InAs/GaAs. Which is actually the working interface? Data about energy barrier height and thermal escape evaluation should be provided.

Another important issue is the comparison with a reference cell without InAs quantum dots which is only mentioned, when discussing EQE variation with IR light (figure 2). All the spectra shown in the manuscript must also show the behaviour of this reference cell for direct comparison.

Again, figure 2 does not present any discussion of the decreasing trend of delta EQE for decreasing wavelength (lower than 680 nm). The origin of this dramatic decrease up to a negative value should be analyzed.

I have some concerns about the theoretical prediction of the conversion efficiency for the proposed cell. The band profiling is extremely simplified with a two level system, E(AlGaAs) and E2. The introduction of InAs QDs actually complicates this profile, by adding QD confined states as well as the WL. These features are neglected by the authors.

Finally, authors declare that the device used in the study has the purpose only to demonstrate this improvement in TPU, so it is not the optimal device which, as they claim, should exhibit thickness optimization, doping, window layer and A/R coating. However, what are the performances of this "un-optimized" device (FF, efficiency, Voc and Jsc) under standard AM1.5 illumination?

Answers to these issues and major revision accordingly should be provided before reaching a final decision on the paper, in my opinion.

Reviewer #3 (Remarks to the Author):

This paper proposes the exploitation of two photon absorption in a heterojunction as a mechanism of surpassing the Shockley and Queisser efficiency limit for solar cells. In the first part the proposed idea is described. In a second part experimental evidence of the operation principles of such a device are presented.

Despite the originality of the idea, I consider that the exposed theory and experiments are not convincing, especially for a top-quality journal such as Nature Communications. Next, I outline my main concerns about the presented work:

Theory:

1. The dense accumulation of charge (electrons) at the interface seems to be incompatible with the sketched band diagram of Figure 1.
2. The spatial variation of the quasi-Fermi levels, in relation with the proposed current flows, is not discussed. This discussion is crucial for understanding the high V_{oc} of the device (higher than the bandgap of the low-bandgap material).
3. The drift-driven accumulation of electrons at the interface is presented as the enabling factor for long-lived electrons, therefore enhancing two-step photon absorption. At high voltages, close to V_{oc} , the electric field is strongly reduced (flatter bands); therefore, the cause for strong two-step absorption is reduced. This issue is not discussed.

Experimental:

4. Related to comment 2, a proof of concept of the proposed device would be to measure V_{oc} higher than the bandgap of the low-bandgap material (GaAs, 1.4 eV). In this sense, the measured 0.7 V (Figure 4a), with an increase of 1 mV due to the second photon flux, is not sufficient. In addition, it would be unexpected that the measured V_{oc} did not increase following an increase in J_{sc} .
5. It is said that 1300 nm photons (0.95 eV) could not trigger interband transitions in the InAs quantum dots, yet there is no experimental support for such an important statement (for example, some kind of spectroscopy of the quantum dots).
6. Concerning the photocurrent measurements shown in Figure 2. Samples were illuminated first with chopped monochromatic light, secondly adding a continuous-wave 1300 nm light source. The power density of the continuous-wave light source was equivalent to almost 4 suns and samples were not cooled during the measurements. These conditions cannot exclude thermal effects altering the results.

In conclusion, the submitted manuscript fails to meet some of the publication criteria of Nature Communications, in particular the following two:

- "The data is technically sound"
- "The paper provides strong evidence for its conclusions".

For all these reasons I suggest it is not accepted for publication in Nature Communications.

We deeply appreciate valuable comments from the reviewers. All the issues pointed out by the reviewers have been addressed as follows one-by-one in detail. According to the reviewer's comments, we updated discussions, experimental results, and figures.

Reviewer #1 (Remarks to the Author):

(#1-1) COMMENT:

line 8 and throughout the text - 'transmission loss' term is ambiguous, would 'recombination loss' be more appropriate?

RESPONSE:

'Transmission loss' is also called 'transparency loss' or 'below E_g loss', which is distinguished from 'recombination loss' [Ref. 2]. Transmission loss is caused by below-gap photons passing through a solar cell (SC). Below-band gap photons with energy smaller than the band gap of SC are not absorbed and do not contribute to create carriers. On the other hand, recombination loss is a loss caused by recombination of photo-created carriers, where photons absorbed in SC play an important role. We updated texts in page 2, lines 7–8.

(#1-2) COMMENT:

lines 16-17 --- dramatic increase in the photocurrent, which exceeded the reported value by approximately two orders of magnitude, but also an increase in the photovoltage.

'Did the photocurrent or the change in the photocurrent exceed the reported value?'

RESPONSE:

The additional photocurrent (the change in the photocurrent) caused by TPU exceeded the reported value by approximately two orders of magnitude. We updated texts in page 1, line 16.

(#1-3) COMMENT:

lines 164 and on:

Theoretical prediction of the conversion efficiency of the TPU-SC. Here, we estimate the expected conversion efficiency of the TPU-SC, which is calculated in a detailed valance framework.

What is 'detailed valance framework'. Give more details of the model, its assumptions and approximations. I suggest proofreading it.

RESPONSE:

We owe the reviewer our thanks. The spell of “valance” was incorrect. It is, of course, “balance”. The reviewer’s suggestion may be on this fault. Our calculation in Fig. 5 is based on a framework called “detailed balance” which was originally proposed by Profs. W. Shockley and H. J. Queisser in Ref. 1. As described in Ref. 1, the detailed balance framework is a framework considering a steady state between carrier generation and recombination at the optimum operation point of SC. This model has been widely used to calculate an ideal, maximum conversion efficiency of SC. Here, we ignore nonradiative processes in SC for predicting the ideal limit of the conversion efficiency. The total photon emission flux, N , with the energy range between E_{\min} and E_{\max} is calculated by using the generalised Planck equation incorporating the effect of chemical potential, μ :

$$N(E_{\min}, E_{\max}, T, \mu) = \frac{2\pi}{h^3 c^2} \int_{E_{\min}}^{E_{\max}} \frac{E^2}{\exp\{(E-\mu)/k_b T\}-1} dE, \quad (1)$$

where T is temperature, h is the Planck’s constant, c is the light velocity, k_b is the Boltzmann constant. By using Eq. (1), generation rates of G_{WGS} in wide-gap semiconductor (WGS), G_{NGS} in narrow-gap semiconductor (NGS), and G_{up} for TPU can be expressed by (see Fig. R1):

$$G_{\text{WGS}} = X f_{\text{sun}} N(E_{\text{WGS}}, \infty, T_{\text{sun}}, 0) + (1 - X f_{\text{sun}}) N(E_{\text{WGS}}, \infty, T_{\text{cell}}, 0), \quad (2)$$

$$G_{\text{NGS}} = X f_{\text{sun}} N(E_{\text{NGS}}, E_{\text{WGS}}, T_{\text{sun}}, 0) + (1 - X f_{\text{sun}}) N(E_{\text{NGS}}, E_{\text{WGS}}, T_{\text{cell}}, 0), \quad (3)$$

and

$$G_{\text{up}} = X f_{\text{sun}} N(\Delta E_c, E_{\text{NGS}}, T_{\text{sun}}, 0) + (1 - X f_{\text{sun}}) N(\Delta E_c, E_{\text{NGS}}, T_{\text{cell}}, 0), \quad (4)$$

where X is the solar concentration factor, $f_{\text{sun}} = 2.16 \times 10^{-5}$ is the solid angle of the sun, $T_{\text{sun}} = 6,000$ K is the temperature of sun, E_{WGS} and E_{NGS} are bandgap energies of WGS and NGS, respectively, ΔE_c is the conduction band offset between WGS and NGS, $T_{\text{cell}} = 300$ K is the temperature of the SC. Relation of E_{WGS} , E_{NGS} , ΔE_c , and valence band offset, ΔE_v , is given by:

$$E_{\text{WGS}} = E_{\text{NGS}} + \Delta E_c + \Delta E_v, \quad (10)$$

Each recombination rate is given by:

$$R_{\text{WGS}} = N(E_{\text{WGS}}, \infty, T_{\text{cell}}, \mu_{\text{WGS}}), \quad (5)$$

$$R_{\text{NGS}} = N(E_{\text{NGS}}, E_{\text{WGS}}, T_{\text{cell}}, \mu_{\text{NGS}}), \quad (6)$$

and

$$R_{\text{up}} = N(\Delta E_c, E_{\text{NGS}}, T_{\text{cell}}, \mu_{\text{up}}), \quad (7)$$

where μ_{WGS} and μ_{NGS} are the quasi-Fermi level separation in WGS and NGS, respectively and μ_{up} is the quasi-Fermi level separation due to TPU (see Fig. R1). Here, we take into

account electrons accumulated at the hetero-interface and TPU occurring in the conduction band. Similar thing happens when holes are accumulated at the hetero-interface and TPU occurs in the valence band. In this case, ΔE_c in Eqs. (7) and (8) is replaced by ΔE_v . According to these relations, the total current, J , generated in TPU-SC is obtained by:

$$\frac{J}{q} = G_{WGS} + G_{NGS} - R_{WGS} - R_{NGS}, \quad (8)$$

where q is electronic charge. In TPU-SC, the following current matching condition of TPU must be satisfied:

$$0 = G_{NGS} + G_{up} - R_{NGS} - R_{up}. \quad (9)$$

The output voltage of TPU-SC is given by

$$qV = \mu_{WGS} = \mu_{NGS} + \mu_{up}. \quad (11)$$

Finally, the total electrical power generated in TPU-SC is calculated as a product of VJ and, hence, the expected conversion efficiency can be estimated by VJ divided by the total incident photon energy. Detailed theoretical model and calculated results for the conversion efficiency of TPU-SC will be presented elsewhere.

In this paper, we added several texts describing the fundamental framework of the detailed balanced model in page 15, lines 2–5. Besides, we updated Fig. 5 and page 16, line 1 because we found a bug in our program codes and fixed it. The calculated conversion efficiency has been underestimated in the previous calculations. The updated calculations maintain well the final conclusion.

Figure R1 | Schematic calculation model of TPU-SC. E_{WGS} and E_{NGS} are band gap of WGS and NGS, respectively. μ_{WGS} and μ_{NGS} are the quasi-Fermi level separation in WGS and NGS, respectively and μ_{up} is the quasi-Fermi level separation due to TPU. ΔE_c and ΔE_v are the CB and VB discontinuity, respectively. G_{WGS} and G_{NGS} carrier-generation rates in WGS and NGS, respectively, and G_{up} is the carrier-generation rate due to up-conversion. R_{WGS} , R_{NGS} and R_{up} is the carrier-recombination rates in WGS, NGS, and at the hetero-interface, respectively.

(#1-4) COMMENT:

The references may be expanded to include results on InAs/GaAs QD IBSC published by other groups, for example, by R.S. Goldman, S. Forrest from UMich, Ann Arbor; M.F. Doty and J. Zide from University of Delaware.

RESPONSE:

Thank you very much for indicating relevant, interesting papers. We added the following references in page 2, line 18 to page 3, line 1. Huang *et al.* clarified the contribution of the wetting layer and QD size distribution on EQE signal by comparing the experimental and calculation results [Ref. 10]. In Ref. 14, two-step photon absorption in GaSb / GaAs type-II QD-IBSC has been reported, where the maximum ΔJ_{sc} obtained by irradiating additional IR light with the intensity of 750 W/cm² was estimated approximately 10 nA/cm² at 200 K and becomes smaller than 1 nA/cm² at the temperature above 250 K. The reported value of ΔJ_{sc} was smaller approximately two orders of magnitude than that of our TPU-SC. Sellers *et al.* have proposed a new type SC structure which attempts optical up-conversion in electrically-isolated up-conversion layers [Refs. 30, 31]. The up-conversion SC proposed by Sellers *et al.* is a sort of luminescence coupling; high-energy photons emitted by radiative recombination of up-converted electron and hole in the up-conversion layers are absorbed in a SC stacked on it. Conversely, in our TPU-SC, up-converted electrons are directly collected by the top electrode.

We also added the following references in page 2, line 18 to page 3, line 1, according to the reviewer#2's comment. Gioannini *et al.* of Ref. 11 dealt with conventional InAs/GaAs QD-IBSCs and developed a simulation model of carrier drift and diffusion taking into account thermal-escape effects as well as carrier capture and relaxing processes in QD-IBSC. Reference 12 focused on electron-hole separation in conventional InAs/GaAs QD-IBSC. Here, ΔJ_{sc} has been successfully observed, but, unfortunately, there was no information about the value. Varghese *et al.* of Ref. 13 found that V_{oc} reduction for QD-IBSCs is not due to thermal carrier escape but due to carrier-capture process from CB into IB, and reported that the carrier capture process is mainly caused by the WLS. Conversely, up-converted electrons in Al_{0.3}Ga_{0.7}As of our TPU-SC are not captured by GaAs due to the internal electric field.

(#1-5)COMMENT:

The authors need to explain how their approach is fundamentally different from the one presented in:

Diane G. Sellers, Stephen J. Polly, Yujun Zhong, Seth M. Hubbard, Joshua M. O. Zide, Matthew F. Doty,

IEEE JOURNAL OF PHOTOVOLTAICS, VOL. 5, NO. 1, JANUARY 2015

"New Nanostructured Materials for Efficient Photon Upconversion".

RESPONSE:

As we mentioned already for the comment #1-4, Sellers *et al.* have proposed a new type SC structure which attempts optical up-conversion in electrically-isolated up-conversion layers [Refs. 30, 31]. The up-conversion SC proposed by Sellers *et al.* is a sort of luminescence coupling; high-energy photons emitted by radiative recombination of up-converted electron and hole in the up-conversion layers are absorbed in a SC stacked on it. Conversely, in our TPU-SC, up-converted electrons are directly collected by the top electrode. Thus, the SC structure proposed by Sellers *et al.* is conceptually different from our TPU-SC. We added texts describing the difference in page 3, lines 17 to page 4, lines 3.

Reviewer #2 (Remarks to the Author):

(#2-1) COMMENT:

Even though the obtained enhancement in external quantum efficiency of the investigated device is undoubtedly large, I think that a more detailed discussion on this effect is missing. To be clearer, the improvement is claimed in the abstract and in the manuscript as the best "ever reported" but for supporting this claim, only reference 16, from the same authors and using a similar dot in well structure, is cited. For a journal like Nature Communications I would expect a more extensive discussion on such a claim, with respect to a larger suite of literature results. Relevant and more updated works proposing different approaches and analysing the competition between equilibrium and non-equilibrium charge-transfer processes must be cited in this respect:

RESPONSE:

We agree with your opinion, and thank you very much for presenting the papers. In the revised manuscript, we added the following references in page 2, line 18 to page 3, line 1. Detailed points of each reference are as follows.

Reference 14 reported two-step photon absorption in GaSb / GaAs type-II QD-IBSC, where the maximum ΔJ_{sc} obtained by irradiating additional IR light with the intensity of 750 W/cm² was estimated approximately 10 nA/cm² at 200 K and became smaller than 1 nA/cm² at the temperature above 250 K. The reported value of ΔJ_{sc} was smaller approximately two orders of magnitude than that of our TPU-SC. We quoted Ref. 14 and added texts describing the points mentioned above in Supplementary Section 3. Gioannini *et al.* of Ref. 11 dealt with conventional InAs/GaAs QD-IBSCs and developed a simulation model of carrier drift and

diffusion taking into account thermal-escape effects as well as carrier capture and relaxing processes in QD-IBSC. Reference 12 focused on electron-hole separation in conventional InAs/GaAs QD-IBSC. Here, ΔJ_{sc} has been successfully observed, but, unfortunately, there was no information about the value. Varghese *et al.* of Ref. 13 found that V_{oc} reduction for QD-IBSCs is not due to thermal carrier escape but due to carrier-capture process from CB into IB, and reported that the carrier capture process is mainly caused by the WLs. Conversely, up-converted electrons in $\text{Al}_{0.3}\text{Ga}_{0.7}\text{As}$ of our TPU-SC are not captured by GaAs due to the internal electric field. These refs. 11, 12, and 13 are added in the introduction.

In the updated paper, we compare ΔJ_{sc} with results reported in several references. As shown in Fig. 4a, the maximum ΔJ_{sc} of our TPU-SC was 0.6 mA/cm^2 at the additional 1300-nm LD power density of 320 mW/cm^2 . In Ref. 33, we reported a saturable behaviour of ΔJ_{sc} in IBSC including InAs QDs embedded in $\text{Al}_{0.3}\text{Ga}_{0.7}\text{As}$ / GaAs quantum well. In that study, the maximum ΔJ_{sc} was $0.15 \text{ }\mu\text{A/cm}^2$ when excited by the additional infrared (IR) light with the power density of 56 mW/cm^2 . As the saturated ΔJ_{sc} is proportional to the IR power density, ΔJ_{sc} can be estimated to be $0.86 \text{ }\mu\text{A/cm}^2$ at the IR power density of 320 mW/cm^2 used in our experiment. Elborg *et al.* investigated the voltage dependence of extra photocurrent for the GaAs / $\text{Al}_{0.28}\text{Ga}_{0.72}\text{As}$ QD-IBSC [Ref. 38]. In that literature, ΔJ_{sc} was $0.44 \text{ }\mu\text{A/cm}^2$ when excited at the additional IR power density of $1,400 \text{ mW/cm}^2$. Sellers *et al.* investigated InAs / GaAs QD-IBSC inserting GaP strain-balancing layer between each QD layer [Ref. 39]. Here, the maximum ΔJ_{sc} was $3 \text{ }\mu\text{A/cm}^2$ at the additional IR power density of 300 mW/cm^2 . Thus, ΔJ_{sc} of our TPU-SC exceeded all the reported values by greater than two orders of magnitude. We added these discussions in Supplementary Section 3.

Besides, we added the following references in page 2, line 18 to page 3, line 1, according to the reviewer #1's comment. The authors of Ref. 10 clarified that the contribution of the wetting layer and QD size distribution on EQE signal by comparing the experimental and calculation results. Sellers *et al.* have proposed a new type SC structure which attempts optical up-conversion in electrically-isolated up-conversion layers [Refs. 30, 31]. The up-conversion SC proposed by Sellers *et al.* is a sort of luminescence coupling; high-energy photons emitted by radiative recombination of up-converted electron and hole in the up-conversion layers are absorbed in a SC stacked on it. Conversely, in our TPU-SC, up-converted electrons are directly collected by the top electrode.

(#2-2) COMMENT:

In the introduction, it is stated that the "key factor" at the base of the increase in TPU efficiency is "the spatial potential fluctuations at the heterointerface". It seems to me that this

effect is inferred to the specific potential barrier, not to any fluctuations. This point should be clarified.

RESPONSE:

Electrons excited in GaAs are accumulated at the $\text{Al}_{0.3}\text{Ga}_{0.7}\text{As}$ / GaAs hetero-interface and up-converted into $\text{Al}_{0.3}\text{Ga}_{0.7}\text{As}$ by the second photoexcitation and the internal electric field. TPU itself occurs at the hetero-interface of $\text{Al}_{0.3}\text{Ga}_{0.7}\text{As}$ / GaAs without InAs QDs. In that sense, the TPU effect might be due to the specific potential barrier. It is noted that the interface containing InAs QDs improves the TPU efficiency as shown in Figs. 2b and 2d. The optical selection rule of the intersubband transition of electrons in an ideal two-dimensional structure is forbidden for light irradiating the two-dimensional plane perpendicularly [Ref. 36]. The finite thickness of the accumulation layer relaxes the selection rule, and, moreover, InAs QDs play a role enhancing the TPU efficiency. Generally, it is well known that the electronic wavefunctions in QDs are quantized on all three dimensions, and light of all polarization directions induces intersubband transitions [Ref. 37]. Thus, electrons accumulated at the hetero-interface are easily pumped into the conduction band of $\text{Al}_{0.3}\text{Ga}_{0.7}\text{As}$ by the excitation light irradiating the two-dimensional plane containing QDs perpendicularly. As you pointed out, the spatial fluctuation at the hetero-interface was not appropriate to interpret the above-mentioned role of QDs relaxing the optical selection rule. We amended relevant discussion in the introduction. Thank you very much for your indication.

(#2-3) COMMENT:

All along the text, electrons are said to be accumulated at the hetero-interface AlGaAs/GaAs/InAs/GaAs. Which is actually the working interface? Data about energy barrier height and thermal escape evaluation should be provided.

RESPONSE:

This strongly relates to the last comment #2-2. The interface structure used is complicate. As we mentioned above, TPU itself occurs at the hetero-interface of $\text{Al}_{0.3}\text{Ga}_{0.7}\text{As}$ / GaAs. InAs QDs play a role enhancing the TPU efficiency. According to these results, we believe that the working interface is $\text{Al}_{0.3}\text{Ga}_{0.7}\text{As}$ / GaAs.

Based on a well-known band discontinuity of the $\text{Al}_{0.3}\text{Ga}_{0.7}\text{As}$ / GaAs hetero-interface, the conduction band offset can be estimated to be 220 meV which corresponds to the barrier height for electron [Ref. 30]. Excited electrons in GaAs can be accumulated at the $\text{Al}_{0.3}\text{Ga}_{0.7}\text{As}$ / GaAs hetero-interface with the large potential barriers, though a small number of electrons are thermally pumped out, which reduces the output voltage. Here, we carefully measured the

temperature dependence of the current-voltage characteristics for TPU-SC with InAs QDs. Figure S1 shows the temperature dependence of the current density when irradiated by the 780-nm LD. The current density increases with increasing the temperature. The inset of Fig. S1 indicates the applied bias voltage dependence of the estimated thermal activation energy E_A . E_A monotonically decreases with increasing the electric field because of lowering the effective barrier height at the hetero-interface. E_A shows the maximum of 221 ± 3 meV at 0.02 V as shown in Fig. S1. Conversely, applying higher positive bias voltage weakens the internal electric field significantly and makes flatter the band. As the forward current increases even at the same bias condition with increasing the temperature, the detected photocurrent decreases rapidly with flatten the band. Thereby, E_A decreases and finally becomes negative with increasing the bias voltage. The maximum E_A excellently coincides with the estimated conduction-band discontinuity between $\text{Al}_{0.3}\text{Ga}_{0.7}\text{As}$ and GaAs [Ref. 32].

We added texts describing the thermal escape property in page 6, line 19 to page 7, lines 2. In addition, we added Section 1 and Fig. S1 in Supplementary Information.

Figure S1 | Temperature dependence of the current density when excited by a 780-nm LD. 780-nm photons directly excite *i*-GaAs. Red circles indicate the measured current density at the bias voltage of 0.02 V as a function of the temperature. The dashed line represents the result of the Arrhenius-type fitting. E_A is the estimated thermal activation energy. The inset shows the bias voltage dependence of E_A . E_A becomes maximum at 0.02 V indicated by the red arrow.

(#2-4) COMMENT:

Another important issue is the comparison with a reference cell without InAs quantum dots which is only mentioned, when discussing EQE variation with IR light (figure 2). All the spectra shown in the manuscript must also show the behaviour of this reference cell for direct comparison.

RESPONSE:

Figures 2a and 2b show the EQE and Δ EQE spectra of TPU-SC with InAs QDs which are the same data presented in the previous manuscript. We newly added EQE and Δ EQE spectra of the reference cell without InAs QDs in Figs. 2c and 2d, respectively. The same absorption edges of $\text{Al}_{0.3}\text{Ga}_{0.7}\text{As}$ and GaAs appear in the EQE spectrum measured without the 1,300-nm LD illumination. As shown in Figs. 2a and 2c, the EQE drop observed below the band gap of $\text{Al}_{0.3}\text{Ga}_{0.7}\text{As}$ was significant for TPU-SC with QDs, which is caused by extra carrier recombination in QDs. TPU-SC without QDs, of course, does not show a signal relating to InAs WL at ~ 920 nm. As shown in Fig. 2d, Δ EQE is obviously generated even in TPU-SC with the hetero-interface of $\text{Al}_{0.3}\text{Ga}_{0.7}\text{As}$ / GaAs without InAs QDs. Therefore, the working interface is $\text{Al}_{0.3}\text{Ga}_{0.7}\text{As}$ / GaAs. Comparison between the Δ EQE spectra suggests that the hetero-interface containing InAs QDs improves the TPU efficiency. As we mentioned for the comment #2-2, three-dimensionally confined QD relaxes the optical selection rule, and, therefore, electrons at the $\text{Al}_{0.3}\text{Ga}_{0.7}\text{As}$ / GaAs hetero-interface are easily pumped into the conduction band of $\text{Al}_{0.3}\text{Ga}_{0.7}\text{As}$ by the excitation light irradiating the two-dimensional plane containing QDs perpendicularly. We updated Fig. 2 and relevant texts in page 8, lines 6 to page 9 lines 2.

Figure 2 | EQE spectra obtained with and without IR light and Δ EQE spectra measured at 290 K. (a) and (c) show EQE spectra of TPU-SC with and without InAs QDs, respectively. The black and red lines represent the EQE spectra measured with and without 1300-nm LD illumination, respectively. **(b) and (d)** show Δ EQE spectra of TPU-SC with and without InAs QDs, respectively. Δ EQE is defined as the difference between the EQE signals measured with and without the 1300-nm LD illumination.

(#2-5) COMMENT:

Again, figure 2 does not present any discussion of the decreasing trend of delta EQE for decreasing wavelength (lower than 680 nm). The origin of this dramatic decrease up to a negative value should be analyzed.

RESPONSE:

In our many trials of cell characterization for the same epitaxial wafer, we have confirmed that cell-to-cell variability in the trend of Δ EQE for decreasing wavelength (lower than 680 nm) exists. For example, the following data is one of the different result. This exhibits a positive value in the shorter wavelength region. Though we do not fully understand the reason, we infer that the trend is relating to the top-electrode metal and semiconductor interface. Fortunately, as

the EQE variation with the IR illumination in the wavelength below the band gap of $\text{Al}_{0.3}\text{Ga}_{0.7}\text{As}$ was obvious for all the devices we tested, our discussion regarding TPU is reliable. We updated texts in page 7, lines 17 to page 8, lines 3.

Figure S3 | EQE and Δ EQE for a different TPU-SC with InAs QD.

(#2-6) COMMENT:

I have some concerns about the theoretical prediction of the conversion efficiency for the proposed cell. The band profiling is extremely simplified with a two level system, $E(\text{AlGaAs})$ and E_2 . The introduction of InAs QDs actually complicates this profile, by adding QD confined states as well as the WL. These features are neglected by the authors.

RESPONSE:

As we mentioned above, TPU itself occurs at the $\text{Al}_{0.3}\text{Ga}_{0.7}\text{As} / \text{GaAs}$ hetero-interface, and InAs QDs play a role enhancing the TPU efficiency. In our theoretical estimation of the conversion efficiency, we neglected QD states enhancing the TPU efficiency because we simply assumed a perfect TPU at the hetero-interface. Our theoretical prediction of the conversion efficiency for the proposed TPU-SC is based on several ideal assumptions such as complete optical absorption, TPU, and carrier collection efficiency. Besides, we ignored any nonradiative process. We added texts in page 15, lines 5–8.

(#2-7) COMMENT:

Finally, authors declare that the device used in the study has the purpose only to demonstrate this improvement in TPU, so it is not the optimal device which, as they claim, should exhibit thickness optimization, doping, window layer and A/R coating. However, what are the performances of this "un-optimized" device (FF, efficiency, Voc and Jsc) under standard AM1.5 illumination?

RESPONSE:

The device used in this study was designed and fabricated to demonstrate the TPU phenomena. The current performances of the device un-optimised for SC under the standard AM1.5 illumination were J_{sc} of 10.2 mA/cm², V_{oc} of 0.77 V, FF of 0.38, and efficiency of 3.0%. We believe that these values should not be simply compared with reported results of highly optimised cells. There are three approaches for improving the efficiency in future. First, intrinsic properties of TPU should be improved. Here, the position of the hetero-interface and the material of wide-gap semiconductor become important. Al_{0.3}Ga_{0.7}As is not necessarily the most suitable material for the wide-gap semiconductor. On the other hand, the cell structure is needed to be optimised by each layer thickness, doping, and window layer. Of course, the device performance strongly depends on various processes such as A/R coating and back- and top-electrodes.

Reviewer #3 (Remarks to the Author):

Theory:

(#3-1) COMMENT:

1. The dense accumulation of charge (electrons) at the interface seems to be incompatible with the sketched band diagram of Figure 1.

RESPONSE:

We updated Fig. 1 as follows.

Figure 1 | Schematic band diagram of TPU-SC (a) at the short-circuit condition and (b) at an operating condition. Sunlight irradiates the $\text{Al}_{0.3}\text{Ga}_{0.7}\text{As}$ side. High-energy photons are absorbed in $\text{Al}_{0.3}\text{Ga}_{0.7}\text{As}$, and excited electrons and holes drift in opposite directions towards n -layer and p -layer, respectively. Below-gap photons for $\text{Al}_{0.3}\text{Ga}_{0.7}\text{As}$ excite the InAs QDs and GaAs. Long-lived electrons separated from holes are accumulated at the $\text{Al}_{0.3}\text{Ga}_{0.7}\text{As} / \text{GaAs}$ hetero-interface, inducing a dramatic increase in the TPU current. E_f in Fig. 1a is the Fermi level. E_{fe} and E_{fh} in Fig. 1b indicate the quasi-Fermi levels of electrons and holes, respectively. ΔE_c and ΔE_v are the CB and VB discontinuity, respectively.

(#3-2) COMMENT:

2. The spatial variation of the quasi-Fermi levels, in relation with the proposed current flows, is not discussed. This discussion is crucial for understanding the high V_{oc} of the device (higher than the bandgap of the low-bandgap material).

RESPONSE:

We added an illustration of TPU-SC at the operating condition in Fig. 1b in order to understand the boosted V_{oc} of the device. The output voltage of TPU-SC corresponds to the gap of the quasi-Fermi levels for electrons in $\text{Al}_{0.3}\text{Ga}_{0.7}\text{As}$ and holes in GaAs. We updated relevant texts in page 5, lines 1–3.

(#3-3) COMMENT:

3. The drift-driven accumulation of electrons at the interface is presented as the enabling factor for long-lived electrons, therefore enhancing two-step photon absorption. At high voltages, close to V_{oc} , the electric field is strongly reduced (flatter bands); therefore, the cause for strong two-step absorption is reduced. This issue is not discussed.

RESPONSE:

At high voltages, close to V_{oc} , the electric field, yes, is reduced, and the band diagram becomes flatter. Therefore, the cause for TPU reduces. The electric field at the operating point exhibiting the maximum output power is also not so strong. We need to perform detailed simulations of the band profile at the operating point in order to maintain a moderate internal electric field even at the operating point by controlling the doping profile near the hetero-interface. We added texts discussing this point in page 18, lines 7–13.

Experimental:

(#3-4) COMMENT:

4. Related to comment 2, a proof of concept of the proposed device would be to measure V_{oc} higher than the bandgap of the low-bandgap material (GaAs, 1.4 eV). In this sense, the measured 0.7 V (Figure 4a), with an increase of 1 mV due to the second photon flux, is not sufficient. In addition, it would be unexpected that the measured V_{oc} did not increase following an increase in J_{sc} .

RESPONSE:

We agree with your opinion. The increase of 1 mV is not sufficient as compared with the band gap difference between $\text{Al}_{0.3}\text{Ga}_{0.7}\text{As}$ and GaAs. As given in Eq. (4), ΔV_{oc} is an increase against $\Delta V_{oc, \text{single}}$ which is the open-circuit voltage measured at the single-color excitation without the 1,300-nm LD illumination. The increase in ΔV_{oc} includes effect of the voltage boost effect at the hetero-interface, which follows an increase in the extra photocurrent ΔJ_{sc} created by the additional 1,300-nm LD illumination. It is difficult to extract the contribution of the voltage boost effect at the hetero-interface from fitting the curve of ΔV_{oc} in Fig. 4b. However, we have clearly demonstrated a difference between the TPU effect caused by the optical process and the thermal population effect. To confirm the contribution of 1,300-nm LD illumination to ΔV_{oc} , we carefully measured ΔV_{oc} as a function of J_{sc} controlled by the 1,300-nm LD illumination or temperature. The results are summarized in Fig. 4d. The blue circles indicate ΔV_{oc} recorded by changing J_{sc} controlled by temperature. Here, the 1,300-nm LD does not shine the device. With increasing the temperature, J_{sc} increases because of increasing thermal carrier population, and, resultantly, ΔV_{oc} reduces. This is a well-known phenomenon. As the band gap change in this temperature variation is approximately 4.5 meV which is given by 5×10^{-4} eV/K of the temperature dependence of the band gap of GaAs, the observed change in ΔV_{oc} was almost caused by the thermal carrier population effect. Conversely, when the 1300-nm LD with the excitation power density of 300 mW/cm² illuminates the device at 290 K, ΔV_{oc} slightly

increases, despite increasing J_{sc} similarly. The clear difference between the thermal effect and the TPU by the second photon flux proves the concept of the proposed TPU-SC. We added several texts discussing the voltage boost in page 12, lines 16 to page 13, lines 13 and the following figure as Fig. 4d.

Figure 4 | Two-step photon up-conversion current at biased conditions. (d) Comparison of the ΔJ_{sc} - ΔV_{oc} relation between increasing the 1,300-nm LD power density and increasing the temperature. ΔJ_{sc} and ΔV_{oc} in Fig. 4d indicate the difference of J_{sc} and V_{oc} against the values (black circle) measured by a 780-nm LD with the intensity of 47 mW/cm² at 290 K.

(#3-5) COMMENT:

5. It is said that 1300 nm photons (0.95 eV) could not trigger interband transitions in the InAs quantum dots, yet there is no experimental support for such an important statement (for example, some kind of spectroscopy of the quantum dots).

RESPONSE:

We added a photoluminescence (PL) spectrum measured at 300 K. The wavelength and power density of the excitation laser were 660 nm and 80 mW/cm², respectively. The PL peak appeared at 1,180 nm corresponds to the fundamental state of the QD transition, which is shorter than that of 1,300 nm of the second-excitation laser. Thus, 1,300-nm photons (0.95 eV) could not trigger the interband transition in InAs QDs. We added the PL result in Supplementary Section 2.

Figure S2 | PL spectrum of TPU-SC with InAs QD measured at 300 K.

(#3-6) COMMENT:

6. Concerning the photocurrent measurements shown in Figure 2. Samples were illuminated first with chopped monochromatic light, secondly adding a continuous-wave 1300 nm light source. The power density of the continuous-wave light source was equivalent to almost 4 suns and samples were not cooled during the measurements. These conditions cannot exclude thermal effects altering the results.

RESPONSE:

We completely agree with you. We need to consider contribution of both TPU and thermal population to the device operation. The maximum power density of the 1,300-nm LD used in this study was equivalent to almost 17 suns. Here, we took into account Air Mass 1.5G solar spectrum. TPU requires photon absorption in the wavelength region between 1,180 nm (the fundamental state of InAs QD) and 5,640 nm (CB discontinuity). Thereby, the estimated photon flux consumed by TPU is approximately 1.4×10^{17} photons/cm² at 1 sun. As the 1,300-nm LD with the power density of 360 mW/cm² corresponds to 2.4×10^{18} photons/cm², the equivalent solar concentration in this study becomes 17 suns. As we mentioned for the comment #3-4, we successfully distinguished the difference between the thermal effect and the TPU caused by the second photon flux. We updated the relevant discussion in page 12, lines 16 to page 13, lines 13, and page 18, line 1.

Reviewers' comments:

Reviewer #1 (Remarks to the Author):

Dear Authors,

Your detailed responses to the the reviewers' comments are highly appreciated. The clarity of the resubmitted manuscript has been greatly improved, and several essential references have been included and discussed. Therefore, I recommend that this revised version of the article is accepted for publication in Nature Communications.

Reviewer #2 (Remarks to the Author):

I appreciate the significant effort faced by the authors to improve the manuscript quality.

However, there are still some points which lead me to evaluate the paper as not technologically sound and strongly supported by data as required for Nature Communications.

First of all, I think that a clear band profile with all the energy levels available within the structure must be provided. This would require a more careful spectroscopic data analysis with respect to carrier dynamics. For example EQE or PL as a function of temperature should be analysed by Arrhenius plots to clearly assess inter level processes within the cell structure. Only after a clear picture depicted on available energy states for confined carriers authors could discuss about the AlGaAs/GaAs interface role. Based on the structure scheme in figure 6 (which, in any case, should be presented earlier in the text) QDs are embedded within GaAs and this is ignored in the discussion.

Moreover, I do not understand the difference at high energy between the EQE curve of the proposed cell and the reference one (without InAs quantum dots). It seems that in the short wavelength region without IR additional illumination, these cells behave differently.

Considering the revisions presented by the authors and involving fundamental understanding of the structure effect on device operation, I am sorry I cannot support publication on Nature Communications.

Reviewer #3 (Remarks to the Author):

In this new version the authors have tackled satisfactorily most of the points I raised on the original manuscript.

The increase in voltage is still low taking into account the high IR illumination density (17 suns), but it is valid as proof of the potential of the proposed device now that thermal effects have been discarded.

I do not have further comments on the manuscript and, therefore, my opinion is that it is ready for publication.

I encourage the authors to investigate carefully the impact of positive bias on the TPU effect.

Thank you very much for reviewing our manuscript. We deeply appreciate your valuable, constructive comments and warm encouragement. All the issues pointed out by the reviewers have been addressed as follows one-by-one in detail. In particular, according to the comments of reviewer #2, we conducted PL and EQE experiments pointed out by reviewer #2. All the updated outcomes strongly support the TPU effects at the hetero-interface.

Reviewer #1:

COMMENT:

Dear Authors,

Your detailed responses to the reviewers' comments are highly appreciated. The clarity of the resubmitted manuscript has been greatly improved, and several essential references have been included and discussed. Therefore, I recommend that this revised version of the article is accepted for publication in Nature Communications.

RESPONSE:

We deeply appreciate your careful review and are very happy that you find the potential interest of our manuscript.

Reviewer #2:

(#2-1) COMMENT:

I appreciate the significant effort faced by the authors to improve the manuscript quality. However, there are still some points which lead me to evaluate the paper as not technologically sound and strongly supported by data as required for Nature Communications.

RESPONSE:

We deeply appreciate your constructive comments. We conducted PL and EQE experiments pointed out. We added new spectroscopic data and analysis with respect to carrier dynamics, in particular, carrier excitation process, as follows.

(#2-2) COMMENT:

First of all, I think that a clear band profile with all the energy levels available within the structure must be provided. This would require a more careful spectroscopic data analysis with respect to carrier dynamics. For example EQE or PL as a function of temperature should be analysed by Arrhenius plots to clearly assess inter level processes within the cell structure. Only

after a clear picture depicted on available energy states for confined carriers authors could discuss about the AlGaAs/GaAs interface role. Based on the structure scheme in figure 6 (which, in any case, should be presented earlier in the text) QDs are embedded within GaAs and this is ignored in the discussion.

RESPONSE:

Thank you very much for your constructive and useful comments regarding the band profile influencing on carrier dynamics at the hetero-interface. We conducted PL and EQE measurements as a function of temperature.

Figures S1a and S1b show PL spectra for TPU-SC with InAs QDs measured at various temperatures and the temperature dependence of the integrated PL intensity, respectively. The wavelength and power density of the excitation laser were 784 nm and 2.1 mW/cm², respectively. The PL represents the ground state (GS) transition of InAs QDs. The PL peak shifts with temperature and obeys a well-known Varshni's relationship. The integrated PL intensity decreases with increasing the temperature. The thermal activation energy evaluated from the Arrhenius plot is 244 ± 4 meV which coincides with the conduction-band discontinuity between the GS of InAs QD and the GaAs band edge [Yang, X. -F., Chen, X. -S., Lu, W. & Fu, Y. Effects of shape and strain distribution of quantum dots on optical transition in the quantum dot infrared photodetectors. *Nanoscale Res. Lett.* **3**, 534–539 (2008).].

Figure S1 | PL spectrum for TPU-SC with InAs QDs. (a) PL spectra measured at various temperatures and (b) the temperature dependence of the integrated PL intensity.

The temperature dependence of the PL intensity reflects the change in the recombining carrier density in QDs. That change is caused by the thermal carrier escape from the confined state. The thermally populated carriers will increase the current. We carefully measured the temperature dependence of EQE. Figure S2 shows the EQE spectra for TPU-SC with InAs QDs measured at various temperatures and the temperature dependences of the current density obtained at typical excitation wavelengths. Figure S2 shows the EQE spectra for TPU-SC with InAs QDs measured at various temperatures and the temperature dependences of the current density obtained at typical excitation wavelengths.

Figure S2 | EQE spectra measured at various temperature and the temperature dependence of the current density. (a) shows EQE spectra for TPU-SC with InAs QDs at various temperatures. (b) and (c) show the temperature dependences of the current density at 780 and 912 nm, respectively. 780- and 912- nm photons directly excite *i*-GaAs and the InAs-wetting layer state, respectively. Red circles indicate the measured current density at the bias voltage of 0.02 V as a function of the temperature. The dashed line represents the result of the Arrhenius-type

fitting. E_A is the estimated thermal activation energy.

Figure S2a shows the temperature dependent EQE spectra. The excitation light was produced by a tungsten halogen lamp, passed through a 140-mm single monochromator, and chopped by an optical chopper with a frequency of 800 Hz. The excitation power density depended on the wavelength and the integrated power density was approximately 2 mW/cm^2 , much lower than that of the one-sun solar irradiance of 100 mW/cm^2 . The beam diameter of the monochromatic light was 1.2 mm on the SC surface. The photocurrent was detected by a lock-in amplifier synchronised with the optical chopper. The EQE was defined as the efficiency of the photocurrent generation under monochromatic excitation, namely, the number of electrons collected as the photocurrent normalised by the incident photon flux at each wavelength. At low temperature, the absorption edges of $\text{Al}_{0.3}\text{Ga}_{0.7}\text{As}$ and GaAs are relatively steep owing to the excitonic feature. With increasing the temperature, the absorption edges shift and the below-gap state attributed to the InAs-wetting layer appears gradually. The EQE signal from QDs was very weak and below the detection limit because of the deeper quantised state. Figure S2b shows the temperature dependence of the current density at 780 nm excited by a LD. 780-nm photons directly excite *i*-GaAs. The excitation power density was 47 mW/cm^2 . The current density increases with increasing the temperature. The inset of Fig. S2b indicates the applied bias voltage dependence of the estimated thermal activation energy E_A . E_A monotonically decreases with increasing the electric field because of lowering the effective barrier height at the hetero-interface. E_A shows the maximum of $221 \pm 3 \text{ meV}$ at 0.02 V. Conversely, applying higher positive bias voltage weakens the internal electric field significantly and makes flatter the band. As the forward current increases even at the same bias condition with increasing the temperature, the detected photocurrent decreases rapidly with flatten the band. Thereby, E_A decreases and finally becomes negative with increasing the bias voltage. The maximum E_A excellently coincides with the estimated conduction-band discontinuity between $\text{Al}_{0.3}\text{Ga}_{0.7}\text{As}$ and GaAs [Ref 30]. Besides, we measured the temperature dependence of the current density at 912 nm corresponding to the wetting layer state. Figure S2c shows the results. We recorded the current at the bias of 0.02 V. Here, the excitation light was produced by a supercontinuum laser, passed through a 270-mm single monochromator. The monochromatic excitation-laser line width was 9.6 nm. The EQE line width of the wetting layer state is approximately 15 nm and the temperature drift of the wetting layer state is approximately 2.9 nm, so that we fixed the excitation wavelength in this experiment. The evaluated thermal activation energy was $254 \pm 5 \text{ meV}$.

These experimental results of PL and EQE as a function of temperature clarify thermal carrier population processes occurring at the hetero-interface. Photo-excited *electrons* are thermally

populated from the GaAs edge to the $\text{Al}_{0.3}\text{Ga}_{0.7}\text{As}$ one, from the InAs wetting layer state to the $\text{Al}_{0.3}\text{Ga}_{0.7}\text{As}$ edge, and from the QD GS to the GaAs edge. We did not confirm an obvious change caused by thermal population of holes, suggesting photo-excited holes reach the p -GaAs contact without captured at the hetero-interface. The following Fig. S3 summarizes these results we obtained. This clear picture clarifies available energy states for confined carriers at the hetero-interface of TPU-SC with InAs QDs.

Figure S3 | Energy states for confined carriers at the hetero-interface of TPU-SC with InAs QDs at ~ 300 K.

As shown in Fig. 2, ΔEQE at the InAs QD GS of 1,186 nm is very weak, suggesting that optical absorption in the single InAs QD layer with the in-plane QD density of $\sim 1.0 \times 10^{10} \text{ cm}^{-3}$ is not enough to contribute to the change in the current generation at the QD GS, whereas QDs obviously improves the TPU efficiency as shown in Fig. 2b. It is noted that ΔEQE is generated even in TPU-SC with the hetero-interface of $\text{Al}_{0.3}\text{Ga}_{0.7}\text{As} / \text{GaAs}$ without InAs QDs. Comparison between the ΔEQE spectra suggests that the hetero-interface containing InAs QDs improves the TPU efficiency. The optical selection rule of the intersubband transition of electrons in an ideal two-dimensional structure is forbidden for light irradiating the two-dimensional plane perpendicularly [Ref 36]. The finite thickness of the accumulation layer relaxes the selection rule, and, moreover, InAs QDs play a role enhancing the TPU efficiency. Generally, it is well known that the electronic wavefunctions in QDs are quantised on all three dimensions, and light of all polarization directions induces intersubband transitions [Ref 37]. Thus, electrons at the hetero-interface obey the selection rule modified by QDs and are efficiently pumped into the conduction band of $\text{Al}_{0.3}\text{Ga}_{0.7}\text{As}$ by the 1,300-nm LD illumination.

In our theoretical estimation of the conversion efficiency, we simply assumed a perfect TPU at the hetero-interface in our model.

We combined Supplementary Sections 1 and 2 for barrier height estimation at the hetero-interface of TPU-SC with InAs QDs. We included all the new experimental outcomes in the updated Supplementary Section 1 and updated discussion in page 7, lines 8-14, page 8, lines 5-8, page 8, line 17 to page 9, line 8, and the figure caption of Fig. 1. Furthermore, as you pointed out, we added sentences briefly describing the sample structure earlier in the text in page 4, lines 5-7.

(#2-3) COMMENT:

Moreover, I do not understand the difference at high energy between the EQE curve of the proposed cell and the reference one (without InAs quantum dots). It seems that in the short wavelength region without IR additional illumination, these cells behave differently.

RESPONSE:

As we explained in the last response regarding this comment, we confirmed that cell-to-cell variability in the trend of ΔEQE for decreasing wavelength (lower than 680 nm) exists in our many trials of cell characterization for the same epitaxial wafer. Though we do not fully understand the reason of the variability, we infer that this trend depends on the device process and, in particular, relates to the top-electrode metal and semiconductor interface. However, we have been tackling further experiments of the EQE spectrum and obtained reproducibly convincing results exhibiting a positive value in the shorter wavelength region as shown in Figs. 2a and 2b. We updated Fig. 2 as follows.

Figure 2 | EQE spectra obtained with and without IR light and Δ EQE spectra measured at 290 K. (a) and (c) show EQE spectra of TPU-SC with and without InAs QDs, respectively. The black and red lines represent the EQE spectra measured with and without 1300-nm LD illumination, respectively. **(b) and (d)** show Δ EQE spectra of TPU-SC with and without InAs QDs, respectively. Δ EQE is defined as the difference between the EQE signals measured with and without the 1300-nm LD illumination.

Reviewer #3:

COMMENT:

In this new version the authors have tackled satisfactorily most of the points I raised on the original manuscript.

The increase in voltage is still low taking into account the high IR illumination density (17 suns), but it is valid as proof of the potential of the proposed device now that thermal effects have been discarded.

I do not have further comments on the manuscript and, therefore, my opinion is that it is ready for publication.

I encourage the authors to investigate carefully the impact of positive bias on the TPU effect.

RESPONSE:

Reviewers' comments:

Reviewer #2 (Remarks to the Author):

The authors have made an heavy job in improving data analysis. There are still some overselling points, especially the claims in the abstract related to the increase in the quasi-Fermi gap and the generation of the substantial additional photocurrent in the TPU-SC, resulting in a high conversion efficiency for intermediate-band SCs. Such an increment is actually not reported experimentally but only proposed as a potential effect of a further optimization of this design. I think this is the main limitation for this manuscript to be published in such a relevant journal as Nature Communications. Therefore, a re-focusing of the abstract in this direction is at least required.

A minor concern deals with the usage of sentences such as in line 111 " electrons are thermally populated", which has been used also elsewhere in the text and which must be corrected.

Thank you very much indeed for reviewing our manuscript. We deeply appreciate your valuable, constructive comments. All the issues pointed out by the reviewers have been addressed as follows one-by-one in detail.

Reviewer #2:

COMMENT:

There are still some overselling points, especially the claims in the abstract related to the increase in the quasi-Fermi gap and the generation of the substantial additional photocurrent in the TPU-SC, resulting in a high conversion efficiency for intermediate-band SCs. Such an increment is actually not reported experimentally but only proposed as a potential effect of a further optimization of this design. I think this is the main limitation for this manuscript to be published in such a relevant journal as Nature Communications. Therefore, a re-focusing of the abstract in this direction is at least required.

RESPONSE:

We agree with your opinion. We re-focused the abstract. In this study, we observed a dramatic increase in the additional photocurrent that we have never seen before, which exceeds the reported values by approximately two orders of magnitude. Conversely, for the quasi-Fermi gap we observed, the increase of 1 mV is not sufficient as compared with the band gap difference between $\text{Al}_{0.3}\text{Ga}_{0.7}\text{As}$ and GaAs. As our paper do not report an increment in the conversion efficiency, the last sentence of the abstract is not appropriate. We updated the last part of the abstract according to the Discussion section. We believe that the generation of the substantial additional photocurrent in the TPU-SC is promising for a potential effect of the TPU-SC.

COMMENT:

A minor concern deals with the usage of sentences such as in line 111 " electrons are thermally populated", which has been used also elsewhere in the text and which must be corrected.

RESPONSE:

Thank you very much for your careful review. We corrected relevant sentences to "electrons are thermally excited". Thank you very much again.

We deeply appreciate your careful review and warm encouragement. We would like to tackle investigation of impact of positive bias on the TPU effect in our future work and open a new field of SC based on the TPU effect.